# Large-scale physically accurate modelling of real proton exchange membrane fuel cell with deep learning

Ying Da Wang [1,7], Quentin Meyer [2,7,8] ✉, Kunning Tang[1], James E. McClure [3], Robin T. White[4], Stephen T. Kelly [4], Matthew M. Crawford[5], Francesco Iacoviello [6], Dan J. L. Brett [6], Paul R. Shearing [6], Peyman Mostaghimi[1,8], Chuan Zhao [2,8] ✉ & Ryan T. Armstrong [1,8] ✉

Proton exchange membrane fuel cells, consuming hydrogen and oxygen to generate clean electricity and water, suffer acute liquid water challenges. Accurate liquid water modelling is inherently challenging due to the multi-phase, multi-component, reactive dynamics within multi-scale, multi-layered porous media. In addition, currently inadequate imaging and modelling capabilities are limiting simulations to small areas (<1 mm$^2$) or simplified architectures. Herein, an advancement in water modelling is achieved using X-ray micro-computed tomography, deep learned super-resolution, multi-label segmentation, and direct multi-phase simulation. The resulting image is the most resolved domain (16 mm$^2$ with 700 nm voxel resolution) and the largest direct multi-phase flow simulation of a fuel cell. This generalisable approach unveils multi-scale water clustering and transport mechanisms over large dry and flooded areas in the gas diffusion layer and flow fields, paving the way for next generation proton exchange membrane fuel cells with optimised structures and wettabilities.

Climate change has shifted the focus from fossil fuels towards clean and renewable energy sources, with the hydrogen economy emerging as a worldwide solution. Hydrogen fuel cells, and proton exchange membrane fuel cells (PEMFCs) in particular, are key to this green revolution due to their high energy conversion efficiency and zero-emission operations[1]. PEMFCs, consuming hydrogen and oxygen to generate electricity and water, offer the advantages of a low operating temperature (<80 °C), high-energy density and quick refuelling[2]. They electrochemically convert hydrogen into protons and electrons at the anode, which react with oxygen at the cathode to generate electricity with water as the only by-product. PEMFCs are a multi-scale porous media comprising of a solid electrolyte membrane sandwiched

between nanoporous electrocatalyst on both sides, covered by a microporous layer (MPL), a microporous gas diffusion layer (GDL), and topped by millimetre-scale flow channels (Fig. 1). This multilayered architecture both maximises gas diffusion to the active catalytic sites and minimizes water accumulation in the catalyst layers[3]. The performance of PEMFCs is highly dependent on the diffusion and utilisation of the fuel and oxidant gases at the anode and cathode, and on the efficient management of the water generated at the cathode[4,5]. At high loads, the generated water may saturate the moisture carrying capacity of the oxygen (air) and condense into droplets in the porous media[2]. If not sufficiently removed, liquid water will eventually accumulate in the GDL and MPL, challenging gas diffusion to the active sites and thereby

[1]School of Minerals and Energy Resources Engineering, University of New South Wales, Sydney, NSW 2052, Australia. [2]School of Chemistry, University of New South Wales, Sydney, NSW 2052, Australia. [3]National Security Institute, Virginia Tech, Blacksburg, VA 24061, USA. [4]Carl Zeiss X-ray Microscopy, ZEISS Innovation Center California, Dublin, CA 94568, USA. [5]Fuel Cell Store, Bryan, TX 77807, USA. [6]Electrochemical Innovation Lab, Department of Chemical Engineering, University College London, London WC1E 7JE, UK. [7]These authors contributed equally: Ying Da Wang, Quentin Meyer. [8]These authors jointly supervised this work: Quentin Meyer, Peyman Mostaghimi, Chuan Zhao, Ryan T. Armstrong. ✉e-mail: q.meyer@unsw.edu.au; chuan.zhao@unsw.edu.au; ryan.armstrong@unsw.edu.au

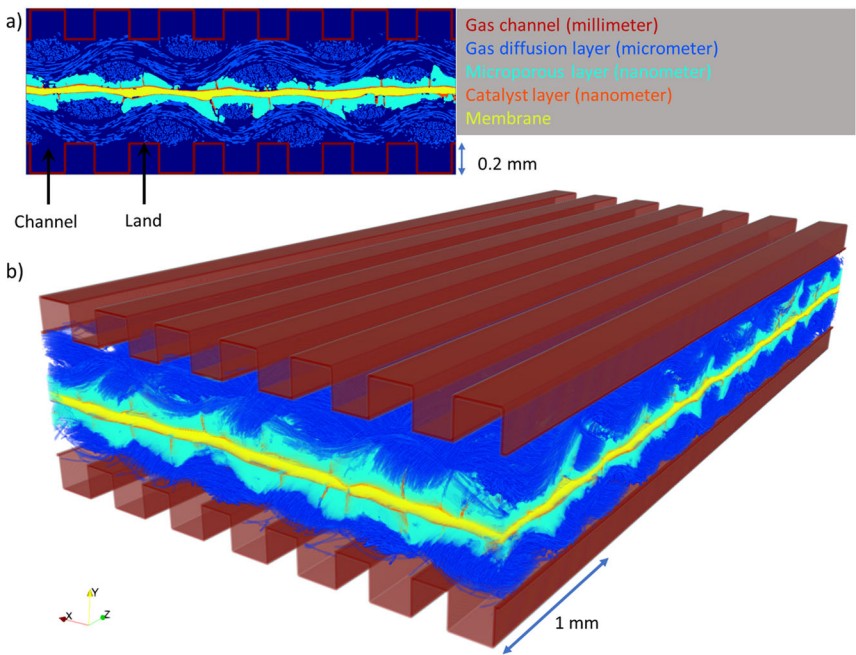

**Fig. 1 | The generated PEMFC domain in this study. a** 2D and **b** 3D rendering of the segmented membrane electrode assembly with artificially overlayed flow channels. The gas channel and land contacting the GDL are labelled.

flooding the PEMFC. Thus, a trade-off between flooding and dehydration is crucial for high-performing PEMFCs[6–8].

Water management in PEMFCs has been improved by modifying the MPL and GDL properties[2,9]. Most notably, large perforations and cracks in the MPL and GDL, initially regarded as manufacturing defects, significantly ease water diffusion from the catalyst layer[10,11]. These features create stable and interconnected water channels inside the electrode, reducing MPL and GDL water content and improving the gas and water balance. This "dual-porosity" flow between MPL/GDL pores and perforations and cracks is well-known for multiphase flow through heterogeneous and fractured porous media such as rocks[12,13], and is critical to optimise PEMFCs multi-phase flow dynamics. Observing PEMFC water transport and diffusion with high levels of detail is essential to studying these spatial variations in porous structure and improving PEMFC designs. Several *operando* water visualisation methods have emerged in PEMFCs such as optical imaging (requiring an opened flow field or transport window), neutron imaging with low spatial and temporal resolution, X-ray radiography (low spatial resolution), and X-ray micro-computed tomography (micro-CT)[14–19].

Of these techniques, X-ray micro-computed tomography (micro-CT) offers the highest spatial resolution (0.5−3 μm), and field of view (1− 6 mm based on a 2000² pixel detector)[20]. In synchrotron facilities, *operando* imaging of PEMFCs can be captured in less than 20 s using micro-CT to capture the liquid water through the MPL and/or GDL pores[2,17,18,21–23]. However, the current resolution and field of view of micro-CT is not able to fully resolve a PEMFC porous structure[3]. Specifically, micro-CT cannot capture the MPL nanopores, while its field of view is restricted to 2−3 gas channels at best[24–26]. Moreover, the GDL fibres (<10 μm in diameter) are poorly resolved by a few voxels causing high aliasing. This trade-off between field of view and resolution limits the capture and analysis of low density features (e.g. water droplets, defects and cracks)[9,15,16,18,27] while maintaining high-resolution, thereby forming an upper hardware limit[28–39] (see Supplementary Table 1a). Beyond-hardware X-ray micro-CT image enhancement was recently introduced using the so-called "super-resolution"[40–43] to circumvent such limitations and improve the resolution of rock samples. In this approach, the resolution of a low-resolution image is enhanced, or super-resolved, to the resolution of a high-resolution small field-of-

view image using convolutional neural networks (CNN)[40]. Although a voxel-wise match is not necessary between the high-resolution and low-resolution images[42], the quality of the super-resolved image is typically higher if the high-resolution image is within the low-resolution one[20]. Such image enhancement using super-resolution convolutional neural networks (SRCNNs) have not been applied to PEMFCs yet.

Numerical modelling of an operating PEMFC or the flow of fluids in its pores and gas channels can be performed in conjunction with *operando* experiments[2,17,18]. Aside from the need for wide field of view and high-resolution, multi-label segmentation of a PEMFC volume is a crucial step for accurate flow modelling over its membrane, catalyst layer, MPL, and GDL. While grey-scale intensities of micro-CT images typically correlate well to the density of the material (X-ray attenuation)[44], the MPL, GDL, and membrane are more challenging to segment due to their similar densities. Machine learning can efficiently capture the contextual information of the spatial correlations and the geometric features, and accurately segment images far beyond traditional segmentation methods[20,42,43,45–47]. Deep learning methods are well-suited and widely used for multi-label segmentation, with well-established use in the micro-CT imaging of porous structures[20,48]. Once the 3D domain is successfully generated, the Navier-Stokes equation can be directly solved on this meshed domain using either the Finite Volume Method with the Volume of Fluid[49,50], or with the Lattice Boltzmann Method (LBM)[51,52], using suitable boundary conditions to mimic the generation and flux of water and gas. Furthermore, electrochemical models may be incorporated into flow modelling at various length scales using direct and agglomeration approaches with co-simulation[53–55]. These approaches allow more efficient, lower spatial-fidelity modelling to be conducted, which can mimic PEMFC *operando* multi-physics dynamics[2,17,18,56].

Such large simulation domains require considerable computational intensity, with even higher requirements for multiphase flow modelling of water and gas transport[52] or even electrochemical modelling of multi-component reactive transport simulations[55]. As such, due to these limits, two paradigms have emerged in modelling PEMFC operating dynamics; *(i)* reduced physics simulations of single or immiscible two-phase flow directly on small subsamples of GDL and MPL porous structures for water management modelling[39,57,58], and *(ii)*

electrochemical simulations with co-simulation to model the operation of a PEMFC with agglomeration techniques to efficiently characterise the multi-layered, multi-scale porous structure[53–55]. In the former case of water management modelling, the highly intensive direct simulation on the voxels of segmented micro-CT images involves a number of simplifying assumptions, for example, since liquid water is generated at the cathode and thus primarily flows from the cathode catalyst layer to the cathode gas channels through the MPL and GDL, a simplifying assumption can be made to limit multiphase flow modelling to the cathode side only[59]. In essence, water removal dynamics in PEMFCs do not require every individual subprocess to be modelled. This is because (i) PEMFC efficiency can be limited by the timescale to remove water; and (ii) the timescale of water removal is dominated by the capillary number (true at length scales of µm to mm), viscosity and density ratio. This limiting behaviour has been observed experimentally using neutron imaging, revealing a direct correlation between liquid water accumulation and voltage losses over a short timescale[60]. The computational resources required to capture water generation and transport dynamics at the cathode are several orders of magnitude higher than single-phase simulations at the anode, which would reach steady state conditions quickly[61]. While $100^3$ to $500^3$ domains can be modelled with a spectrum of workstation-class resources, domains exceeding $1000^3$ require super-computing resources[62], with over 10,000 CPU cores and 100 GPU devices. As a result, direct multi-phase flow simulations of PEMFCs are usually performed on domains in the order of $200^3$ using workstation-class resources[39,57,58,63–66], and focus exclusively on the GDL or MPL. LBM studies on larger PEMFCs derived from micro-CT imaging[56,67] treat water as a solid phase and gas as single-phase flow. More advanced multi-component studies suffer from small fields of view (1.5 mm²) or poor voxel resolution (5 µm) issues[68,69] (see Supplementary Table 1b). Similarly, memory scaling limitations in 3D super-resolution (4 times super-resolution results in 64 times memory increase) have restricted its applications to the generation of small super-resolved cubes for analysis and upscaling[20,70]. Therefore, although large-scale high-resolution imaging and multi-label simulations of PEMFCs are necessary to drive the next technological innovation, the domain size is currently restricted by limitations in both experimental imaging and computational modelling resources.

Herein, both the micro-CT imaging resolution and multi-phase flow modelling capabilities of PEMFCs are significantly advanced by generating an exceptionally large domain of 16 mm² surpassing current hardware limitations (beyond-hardware), and utilising large-scale computing resources (>20,000 CPU cores and 1000 GPU cores) for simulations. A low-resolution, low-quality image of $275 \times 1000 \times 2000$ voxels at 2.8 µm is super-resolved in 3D by an innovative CNN architecture to $1100 \times 4000 \times 8000$ voxels at 700 nm, combining the upper limits of both resolution and field of view. The membrane, catalyst layer, MPL, GDL, pore space, and gas channels are then segmented by a CNN. The resulting image is used to model water and gas transport using single and multi-phase LBM on supercomputing clusters. This improvement represents several orders of magnitude of advancement in the imaging and modelling capabilities of PEMFCs (see Supplementary Table 1a and 1b). Furthermore, such a large and high-resolution domain allows one to investigate the heterogeneous distribution of micro-features over several millimetres such as cracks and defects in the MPL, holes in the carbon paper weave, and alignment and misalignment of gas channels over the GDL[71]. Finally, this study cements that the significant structural and wettability factors across these length scales must be studied by such large PEMFC samples.

## Results

### Super-resolution

The super-resolution algorithm (DualEDSR), developed specifically for this study to handle large 3D images efficiently, is trained on the high- and low-resolution registered images. This involves firstly imaging the whole domain at a low-resolution, and then imaging a small sub-domain at high-resolution with a region of interest scan[72]. The corresponding sub-domain within the low-resolution image is used with the high-resolution image to train DualEDSR to generate super-resolved images from other unseen low-resolution images. DualEDSR is outlined in full detail in "Methods: Super-resolution", and comprises a pair of 2D EDSR networks[73] trained in tandem to efficiently super-resolve the X-Y and Z directions, facilitating the practical super-resolution of large-scale images. The overall validation accuracy as measured by the Peak Signal to Noise Ratio on a unseen section of this sub-domain is 31 dB which is <0.1% mean squared error. This improvement in image resolution and quality to the high-resolution domain with reduced noise greatly improves the accuracy of the segmentation in later steps. A visual comparison of the low-, high- and super-resolution images from the validation set can be found in Supplementary Fig. 1 alongside detailed performance comparison with 3D-EDSR[73]. Briefly, DualEDSR trains five times faster than 3D-EDSR while achieving a similar accuracy. When super-resolving a $150 \times 150 \times 225$ block, 3D-EDSR consumes over 700 GB of memory in the final 3D convolutional layers and needs CPU resources with pagefiling, taking over 2 h. On the other hand, DualEDSR consumes <1GB, and successfully generates the super-resolved $600 \times 600 \times 900$ block in under 1 min, making DualEDSR a superior approach for large 3D domains.

The low-resolution image of the physical PEMFC sample acquired by micro-CT (details in the "Methods: X-ray micro-computed tomography of proton exchange membrane fuel cell" section) measures with $275 \times 1000 \times 2000$ voxels and 2.8 µm voxel resolution (Fig. 2a, b) and represents the hardware field of view limit of the Zeiss Xradia Versa 620 X-ray Microscope used in this study, and this image is super-resolved using DualEDSR. The resulting super-resolved image has $1100 \times 4000 \times 8000$ voxels with 700 nm voxel resolution (Fig. 2c) and 3D render in Supplementary Fig. 2. This represents the upper hardware limit of voxel resolution (700 nm) for the particular detector used in this study and is effectively combined with the wide field of view. The resolution of the carbon fibres is particularly enhanced, with individual fibres easily distinguishable in the super-resolved image. Furthermore, significant improvements in the MPL/GDL and catalyst layer/MPL interfaces are achieved. Finally, the catalyst layer and electrolyte membrane are far easier to distinguish from surrounding materials, while the background noise has been significantly reduced as well. A close-up can be found in Supplementary Fig. 5. Only minor inaccuracies are sparsely observed, caused by artifacts in the low-resolution image (random contrast and noise variation) (Supplementary Fig. 3). In terms of time efficiency, training DualEDSR took under 12 h to reach a plateau on a single RTX Titan GPU (see Supplementary Fig. 1), and generation of the super-resolved $1100 \times 4000 \times 8000$ voxels @ 700 nm voxel resolution image took under 1 h of GPU time. In comparison, if one were to attempt to generate this high-resolution, wide field-of-view image by zoom-in and stitching regions of interest, the single small high-resolution training block (Fig. 6) took ~11 h to acquire, for a field of view of $600 \times 600 \times 900$ voxels @ 700 nm resolution. To collect high-resolution data across the entire sample would require approximately and at minimum (assuming no overlap of the data sets) 108 such high-resolution blocks, totalling 1188 h of data collection (7.5 months at 8 h a day, 5 days a week), not including data set stitching or allowing for needed overlaps to ensure proper alignment of the data sets. This is an image acquisition time reduction of at least 1 order of magnitude in the most ideal case. Furthermore, once DualEDSR is trained on a specific type of material and imaging condition, the procedure can be repeated. Training a new DualEDSR for other samples and imaging conditions is similarly straight-forward by generating new low-resolution and high-resolution images as described in the "Methods: X-ray micro-computed tomography of proton exchange membrane fuel cell" section. Overall, this significant resolution improvement allows one to overcome the

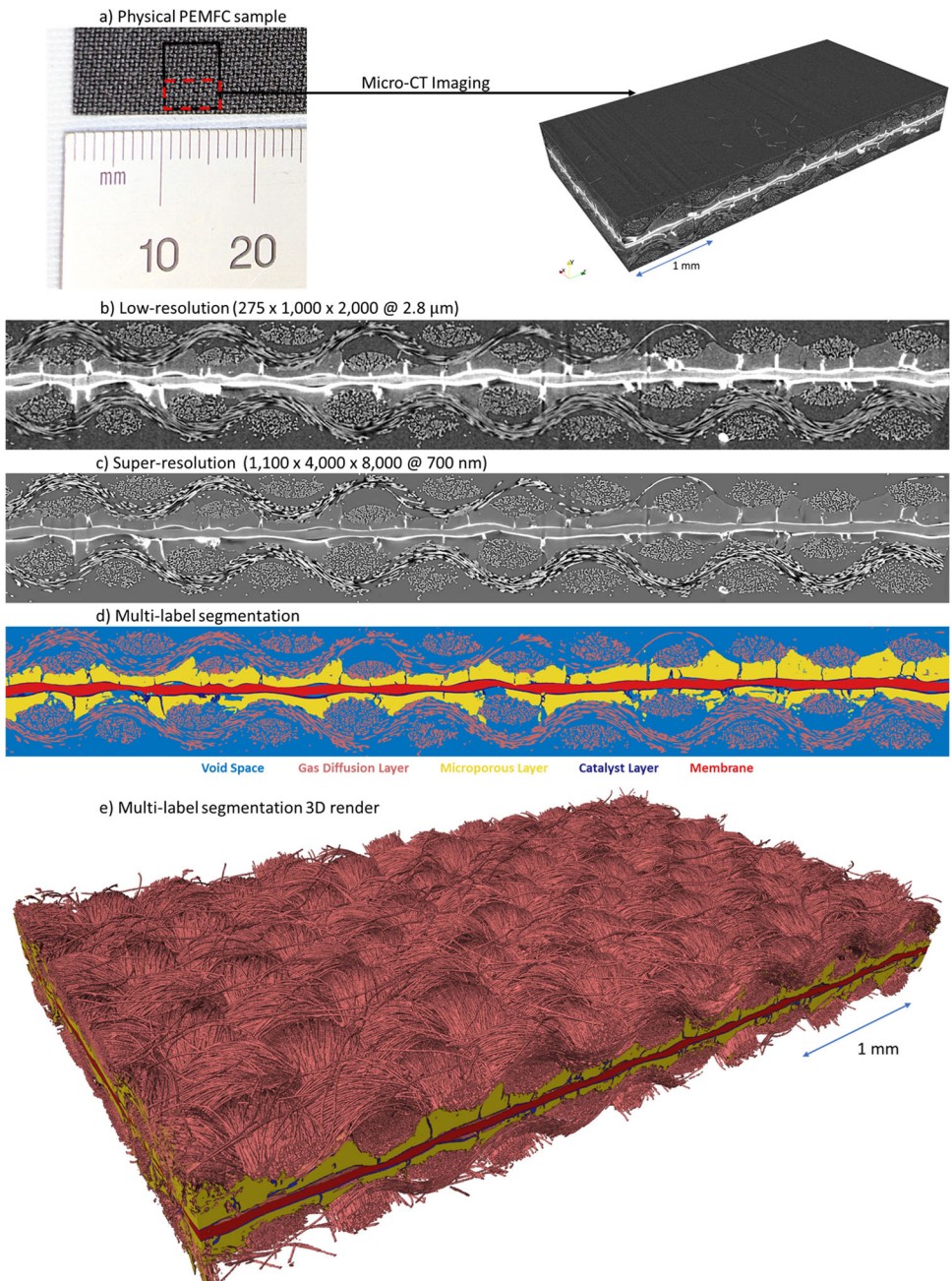

**Fig. 2 | Micro-CT imaging, deep learned super-resolution and multi-label segmentation. a** Photograph and micro-CT image of the PEMFC physical sample. **b** 2D cross section of the low-resolution (275 × 1000 × 2000 at 2.8 μm) for comparison with **c** 2D cross section of the super-resolution (1100 × 4000 × 8000 at 700 nm), and **d** the multi-label segmentation with **e** a 3D render of the segmented super-resolved domain.

hardware-based trade-off between resolution and field of view, which will be invaluable to improve the quality of the multi-label segmentation and following multi-phase flow model, and represents the initial major advancement of this work.

## Multi-label segmentation

Full feature PEMFC segmentation was performed using the workflow and methodology described in the "Methods: Full feature segmentation" section involving the generation of ground truth using machine learning and the training of a CNN to segment the entire domain, which is an established and generalisable segmentation methodology[46,47]. During training and testing, the training accuracy and testing accuracy reached 97.3% and 96.2% after 100 epochs.

Training epochs were monitored to identify when the testing loss remained unchanged after several decreases in the learning rate while training loss continued to decrease (Supplementary Fig. 4a). The confusion matrix (Supplementary Fig. 4b) of the labelling accuracy indicates a high segmentation accuracy (>93%) for the void, fibre, MPLs, and catalyst layers phases, and an accuracy of 86% for the membrane. A representative slice of this segmentation is shown in Fig. 2d with a full 3D render in Fig. 2e. For the subsequent heterogeneity analysis and flow simulation, the perpendicular fibre and parallel fiber phases are merged into a single fiber phase. A close-up view can be found in Supplementary Fig. 5 and the visualised renders of individual PEMFC layers of the full feature segmentation can be found in Supplementary Fig. 7.

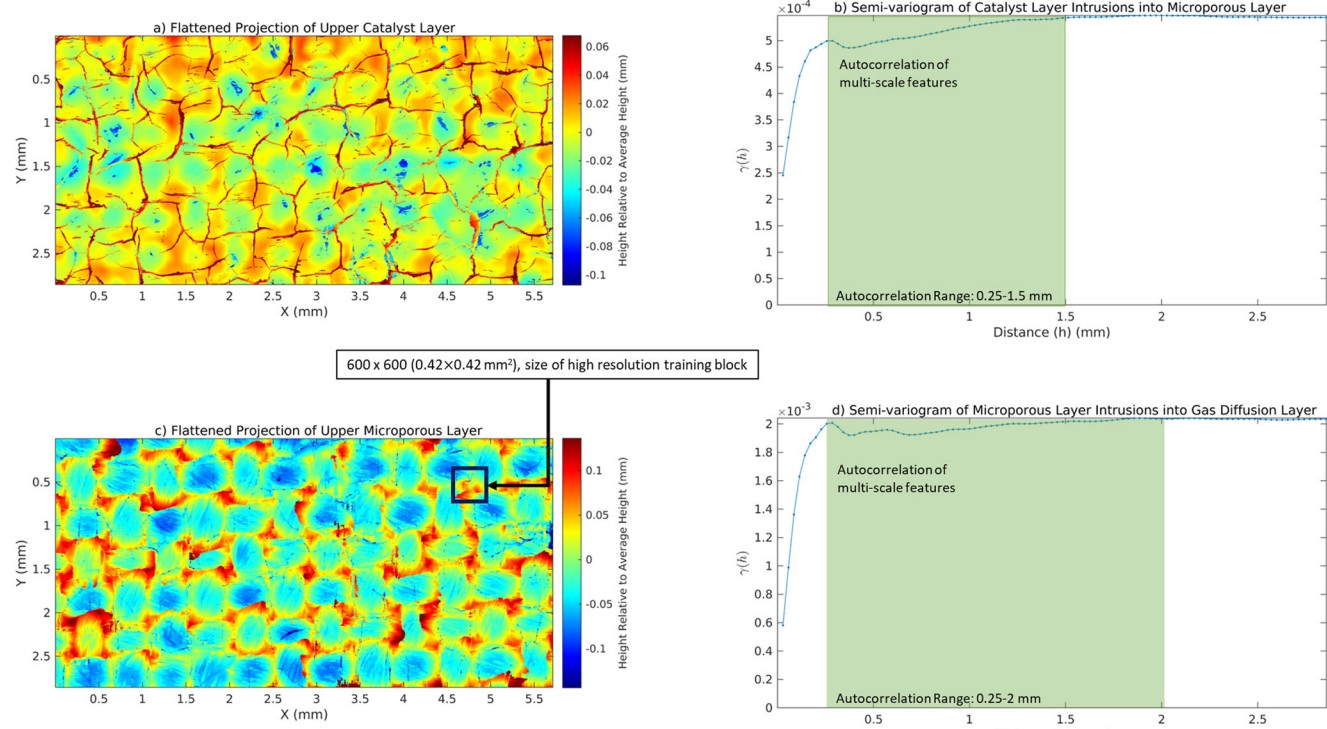

**Fig. 3 | Spatial analysis of the super-resolved, multi-label segmented image.** **a** 2D projections of the catalyst layer intrusions into the MPL fractures and **b** semi-variogram with typical length between intrusions of up to 1.5 mm. **c** 2D projections of the MPL pinching by GDL and intrusion into GDL weave holes and **d** semi-variogram with an autocorrelation range of 0.25–2 mm.

Furthermore, to emphasise the importance of the deep learned super-resolution and multi-label segmentation, attempts were made to segment the low-resolution image using manual segmentation (Avizo software, without deep learning) and with multi-label CNN segmentation of the low-resolution dataset. In the case of manual segmentation on the low-resolution image, the MPL was indistinguishable from the GDL and the layer failed to be segmented. In both cases, the thicknesses of the catalyst layer and GDL fibres was oversegmented by a factor of 3 or more due to image blur and lack of spatial features, resulting in excessive contact area between GDL fibres and overestimation of catalyst deposition thickness. A visual comparison between the segmentation of the low-resolution image using *(i)* manual segmentation with Avizo software (Fisher Scientific) and *(ii)* multi-label CNN segmentation as outlined above is given with a further comparison with *(iii)* the super-resolved multi-label CNN segmentation and can be found in Supplementary Fig. 6. These excessive physical inaccuracies in the pore structure preclude the possibility of accurate flow simulations on the low-resolution domain, with or without deep learned segmentation.

**Heterogeneity analysis**
Following the deposition of the MPL slurry onto the woven GDL, cracks appear in MPL regions with weaker woven support[74]. As the catalyst layer is coated on the MPL, it may penetrate these cracks as well. These cracks, or defects, have a significant role in the cell performance and flow dynamics, creating easier pathways for gas and water (rendered in Supplementary Fig. 9). These defects are isolated here by generating a heatmap and semi-variogram of the thickness of the catalyst layer and MPL from the full-size super-resolved and multi-label segmented image obtained prior and depicted in Fig. 2d–e. The flattened projections of the catalyst layer and the GDL as well as the autocorrelation of these projections can be seen in Fig. 3, with flattened projections shown as relative to the average height of each layer. Two sills in the catalyst layer semi-variogram are detected, which correspond to micrometre scale variations along the membrane surface (100–300 μm), and millimetre scale variations in the MPL fracture network (1.5 mm). Similarly, the MPL intrusion map shows a sill-range value exceeding 0.25 mm, with variation up to 2 mm, caused by pinched and intruded regions. Overall, this analysis demonstrates that such wide fields of view are necessary to capture these defects as smaller domains do not provide an accurate representation[75]. This study's domain measures equivalently $3600^3$ with gas channels imposed, which is orders of magnitude higher than the typical imaged and synthetic domains for flow modelling studies ($100^3$ [63,64], at the exception of a single-phase study on a $650^3$ domain[56]).

**Permeability and velocity field heterogeneity**
Single-phase analysis is performed in order to *(i)* probe the required computational resources for such a simulation on this large 3D dataset of a super-resolved, multi-label segmented PEMFC image, *(ii)* identify regions of higher flow within and around the GDL, and *(iii)* determine if super-resolution provides a sufficient resolution for flow simulations in this PEMFC. The permeability and flow field of the entire PEMFC is determined using single-phase LBM as outlined in the "Methods: Direct flow simulation" section. This represents the flow of gas through both sides of the PEMFC. As the MPL is unresolved, it is treated as a solid, with flow through the GDL and the larger gaps and openings in the GDL weave. As the image does not inherently contain a gas channel, and this single-phase simulation does not include a water phase, the domain is closed-off by flat walls.

This domain was simulated with 20,736 CPU cores (Gadi Supercomputing Cluster), took 66,000 timesteps to converge (8 h of walltime), and required 13 TB of RAM. While such a computing requirement may represent a bottleneck in the direct simulation modelling of large-scale PEMFC, these are anticipated to alleviate over time, since LBM scales well with parallelisation. A normalised render of the velocity field in Fig. 4a reveals that the flow is four orders of magnitude higher in and around the GDL weave, which is consistent

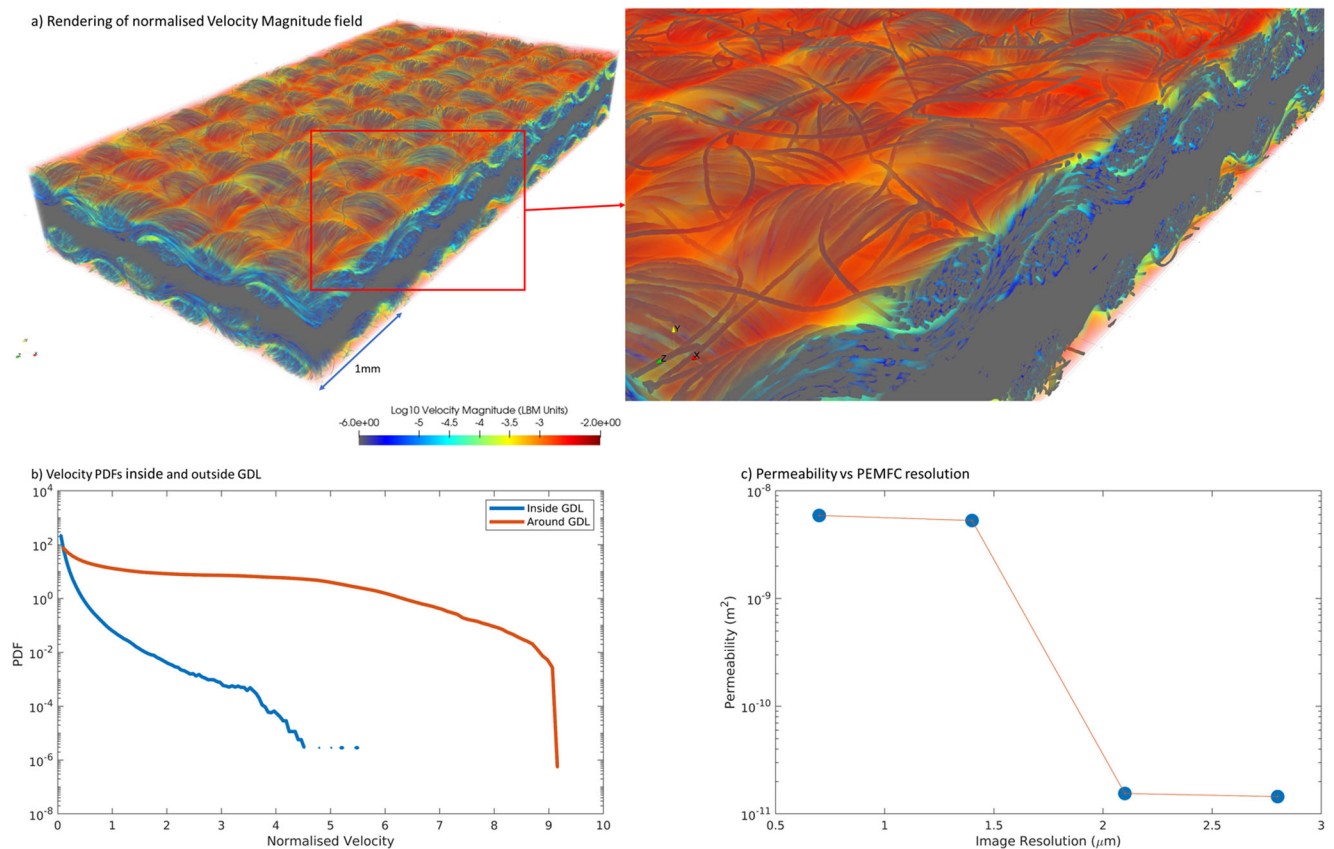

**Fig. 4 | Velocity field distributions within the PEMFC. a** Fully resolved velocity magnitude field of the PEMFC image at 700 nm voxel resolution. **b** Velocity PDFs $\frac{|\bar{v}|}{|\frac{1}{V}\int_V \bar{v}dV|}$ of flow within and around the GDL. **c** Analysis of permeability vs image resolution of the PEMFC.

with the geometry of the material itself. Therefore, any water exiting the MPL will preferentially be wicked away along the holes and inter-weave bundle channels of the GDL.

Since this PEMFC contains multi-scale features, a flow simulation of the domain and the permeability value reported as the bulk average of the domain do not represent a single porous feature of the PEMFC. Thus, the velocity field distribution identifies the flow contributions within the porous internal structure of the GDL and around the GDL. This gives an indication of the predominance of the advection domi-nated flow in the open space and the inter-weave holes transitions and of the mixed advection-diffusion flow in the GDL pore space. The difference in flow within and around the GDL is unveiled using velocity Probability Density Functions (Fig. 4b, calculated by $\frac{|\bar{v}|}{|\frac{1}{V}\int_V \bar{v}dV|}$). The regions inside and surrounding the GDL were isolated using a morphological closing operation with a spherical radius of 14 voxels. While cross-sections and coarsening analysis of the flow pathways could be performed, these require a sensitivity analysis of the different domains and flow conditions to generate results with physical meaning.

Direct flow simulation requires a well-resolved pore space to accurately model flow in the domain, which is only achievable here using the super-resolved domains[76]. In the final paragraph of the "Results: Multi-label segmentation" section, attempts to segment the low-resolution domain already showed highly inaccurate pore struc-tures for flow simulation. The influence of the image resolution on single-phase flow in porous media is further revealed by down-sampling the segmented super-resolved image. While these smaller domains are more efficiently modelled, the pore space increasingly closes off near pore space constrictions and fiber contact points, causing inaccurate no-flow regions in what would otherwise be open

flow paths. Figure 4c shows the computed permeability over the downsampling factors, with zoomed-in areas of the pore-space detail for each down-sampling level. While the super-resolved PEMFC retains a reasonable level of detail when downsampled by a factor of 2 (1.4 μm resolution), further downsampling reduces the permeability from $5.9 \times 10^{-8}$ m$^2$ to $1.45 \times 10^{-11}$ m$^2$, with a percolation threshold between 1.4 and 2.1 μm. Additional visual representations of the velocity can be found in Supplementary Figs. 10 and 11.

The simulation demonstrates the analysis capabilities over several length-scales on a single domain using wide field of view and high-resolution imaging. Furthermore, this technological advancement allows one to elucidate the water transport and gas diffusion in PEMFCs in steady-state and flooded conditions.

## Water transport and gas diffusion modelling

The generation and transport of water in the large-scale PEMFC is modelled using direct multi-phase flow simulation. The super-resolution of the PEMFC image provides a representation of the pore space of the GDL while the wide field of view allows to simulate water transport and removal through the MPL fractures and GDL inter-weave holes as outlined in the "Methods: Direct flow simulation" section. The "Results: Multi-label segmentation" and "Results: Permeability and velocity field heterogeneity" sections show that segmenting the low-resolution domain or downsampling the super-resolved domain will result in geometries unsuitable for flow simulation. As such, the full super-resolved, multi-label segmented domain is used. Individual contact angles and other material specific properties may be assigned to each layer following the multi-label segmentation. As the water is generated at the cathode, this model solely considers the upper half of the cell. This assumed that limited back diffusion occurs when using a weaved gas diffusion layer and relatively uniform MPLs[59]. While some

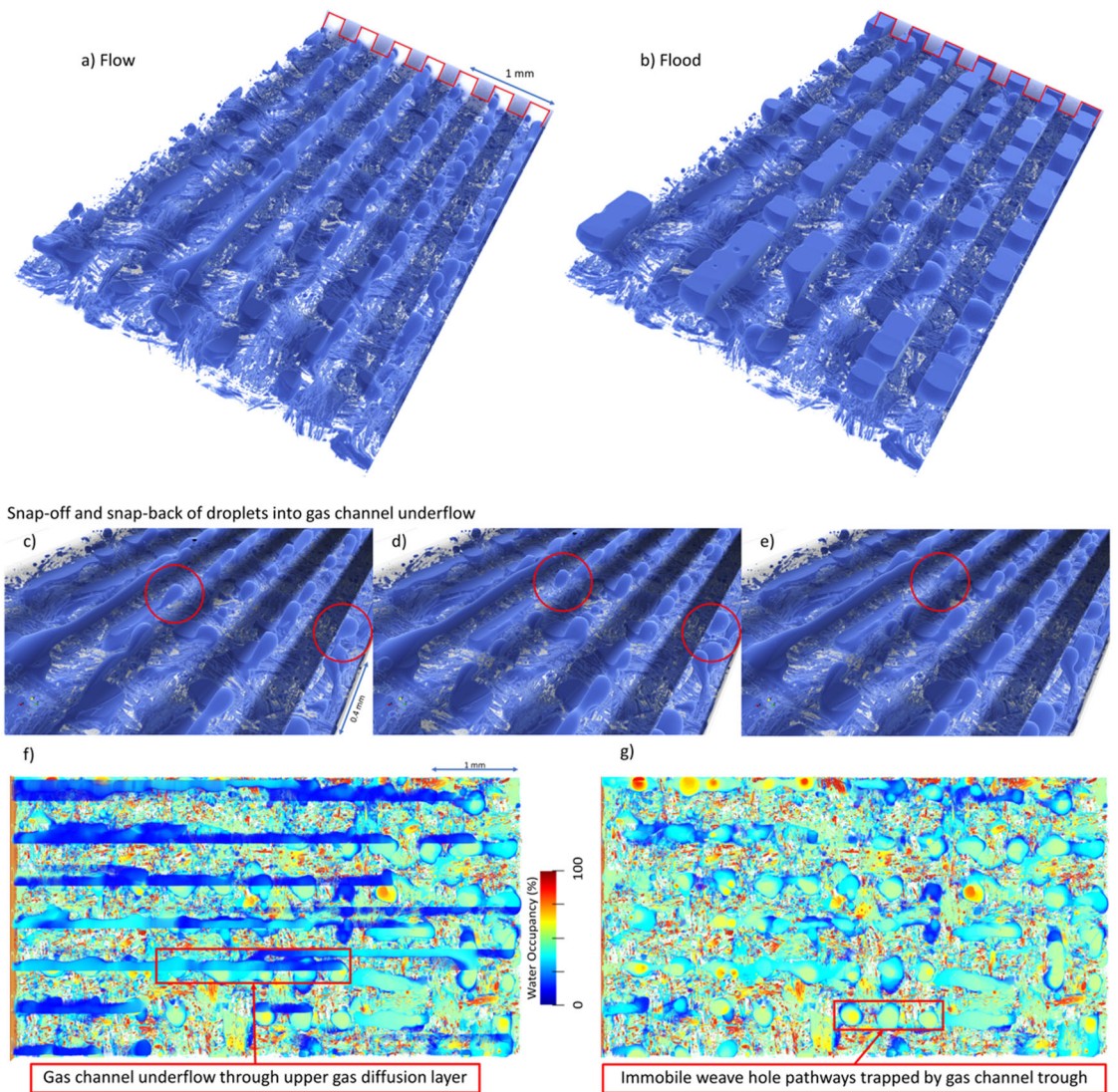

**Fig. 5 | Water-air flow simulations in the PEMFC.** Flow patterns for simulated **a** flow, and **b** flood regimes. **c**–**e** Flow patterns showing droplets snapping out of and back into GDL weave holes in gas channel underflow. **f**–**g** Water occupancy obtained by averaging over timesteps, showing water flow interactions between GDL and gas channels. **f** Includes gas channels, and **g** shows only up to the bottom of the gas channels.

water will be present at the anode, it is secondary to the cathode in terms of management of its removal, which is where it is rate-limited. Parallel gas channels of 0.2 mm width and depth are overlayed over 30% of the segmented GDL thickness to emulate a reasonably compressed PEMFC[77,78] (Fig. 1).

For this simulation, the contact angles for all layers was set to 120 degrees to model hydrophobic carbon material[79] as this sample does not contain PTFE, which would result in more complex mixed wetting surface distributions. The inlet flow rate Reynolds number heavily influences the two-phase flow simulations. The Reynolds (Re) number is calculated as $Re = \frac{\rho_{air} v_{inlet} H}{\mu_{air}}$ where $H$ is the thickness of the PEMFC. The simulation is performed with the injection of air (vapour) through the PEMFC gas channels with a Reynolds number of 1. An air to liquid water volumetric ratio of 9:1 was set on the surface of the MPL as described in the "Results: Direct flow simulation" section to mimic non-flooded Buckley-Leverett[80] displacement of condensed water within the MPL. To emulate the convective force of a minor temperature difference between the MPL and gas channel, an upwards body force of $10^{-6}$ (lattice units) was applied—similar to previous water management studies that impose pressure or velocity conditions in lieu of explicitly modelling temperature fields[64,65]. This assumes that the effect of

temperature gradient and any other upwards diving forces generated by density differences are relatively homogeneous, and in this case, results in a capillary number of $10^{-5}$ in the upwards direction with in the GDL. This flow regime was simulated until the water saturation reached a steady plateau at 22%, after water droplets were transported over one length of the sample. Water flooding of the PEMFC was then simulated by halting the gas flow (Re of 0) until the water content reached 75%[81]. The PEMFC was then purged by setting the flow rate to its previous value, with the PEMFC the water content returning to 22%. This model required 30,000,000 timesteps on the Summit supercomputer. A 3D rendering of flow, and flood modes is shown in Fig. 5a–e. These reveal that water predominantly accumulates near weave holes and under the lands (see animated video with flow-flood-purge cycle in Supplementary Video 1). This simulation domain and routine overcomes the size limitations of previous large-scale studies by two orders of magnitude[56], and successfully models multi-phase flow dynamics.

Water droplets are removed efficiently when the GDL holes are well-aligned with the gas channels, while they remain trapped when the holes are facing the lands. Furthermore, water does not diffuse into the hydrophobic weave fibres as the transport resistance through the holes is lower. This is particularly visible in close-up of the GDL and gas

channel regions (Fig. 5c−e), with water droplets initially trapped under the lands, wicked away from a GDL hole into the channels, and finally re-attached to larger droplets trapped under GDL holes−causing partial flooding. Heat maps of the water flow (generated by summing and normalising the water distribution over the simulation frames) in flow mode (Fig. 5f, g) reveal (f) localised flooding and underflow between GDL weave holes and gas channels, and (g) regions with higher water retention in the holes within the GDL weaves that lead to the land region of the flow field. These findings greatly improve the understanding of the water transport and diffusion across lands and channels, demonstrating complex interactions. Furthermore, these findings suggest that GDL and flow fields should be designed simultaneously, as even a slight misalignment of the weave holes and land such as the one observed here creates localised water flooding with extensive water retention in steady-state conditions. These complex design considerations between water management and PEMFC porous structure can also be probed with low-resolution *operando* experiments such as 2D neutron imaging and X-ray radiography[34,35,82]. These would supplement the numerical modelling of the super resolved images and could be used to tune and validate water management modelling for sensitivity studies.

Overall, the hydrophobic and porous GDL creates two independent flow regimes, with (i) water flowing upwards through higher porosity pathways through the GDL, then from the GDL weave holes into the gas channels, and (ii) gas diffusing through intra-fiber regions of the GDL into the MPL and catalyst layers. Although not investigated in-depth here, the MPL fractures should create preferential liquid water pathways, while the homogeneous regions facilitate the mass transfer of gases. While these simulations considered here uniformly hydrophobic MPL and GDL, local heterogeneities may significantly influence the gas and water pathway and could be investigated further using a similar approach. Finally, the multi-phase flow model could be further enriched using electrochemical modelling considering heterogeneous current densities and heat generation[53–55].

## Discussion

This study greatly advances the understanding of liquid water transport in PEMFC cathodes, unveiling water droplets clustering and accumulating over channels and lands with a 700 nm voxel resolution over a 16 mm$^2$ area. This is achieved by modelling on a high-fidelity image of the PEMFC, generated by deep-learned super-resolution and multi-label segmentation. This approach provides an in-depth analysis of gas and water transport over the MPL, GDL, and gas channels simultaneously at maximum spatial fidelity and extent without volume averaged up-scaling. This beyond-hardware approach reveals water clustering from small features (<100 μm) and growth (>1 mm) and overcomes the typical imaging field of view and resolution trade-off conventionally restricting the analysis of such volumes. As water transport and management in PEMFCs is strongly impacted by the architecture of this complex microporous media, improving the digital characterisation of these structures provides the necessary level of spatial resolution to study significant structural and wettability-based design considerations. Such multi-scale and wide-area considerations are evidenced in this primary study, suggesting the potential benefits of aligning the catalyst intrusions, MPL fractures, GDL weave holes, and gas channel to enhance water management. As such, through sensitivity analysis of large-scale PEMFC domains, an improved understanding of the porous structure influences on gas-water flow in PEMFCs will lead to more effective designs. These beyond-hardware imaging and modelling findings extend past the fuel cell field to achieve higher resolution imaging of larger fields of view than previously practical. Applications span from enhanced sample inspection for quantitative analysis to direct modelling and *operando* experiments where high spatial detail is essential.

## Methods

### X-ray micro-computed tomography of proton exchange membrane fuel cell

A commercially prepared 25 cm$^2$ membrane electrode assembly composed of two woven carbon fiber cloth GDLs with hydrophobic MPLs (410 μm thick, W1S1011, CeTech), and catalyst layers (0.5 mg$_{Pt}$ cm$^{-2}$, 60% Pt/C on Vulcan) sandwiching a perfluorosulfonic acid proton exchange membrane (50.8 μm thick, Nafion NR-212, Dupont) was used for this work (FuelCellStore, US). A sample (6 mm × 7 mm) of the membrane electrode assembly was probed with a razor blade and mounted on a graphite rod for X-ray computed tomography imaging. A laboratory X-ray CT system (ZEISS Xradia 620 Versa X-ray microscope, ZEISS Innovation Center California, US) was used for 3D imaging of the membrane electrode assembly. A source voltage of 60 kV and source power of 6.5 W were used with a low energy filter (LE2) for all samples. All samples were acquired using the Scout-and-Scan Instrument Control System software (ZEISS). The full field of view measuring 6 mm × 3 mm × 0.8 mm (Fig. 2a) of the sample was imaged on the 0.4X objective, using 1601 projections and a 2 s exposure time using vertical stitching. By positioning the source 1.25 cm from the sample, and the detector 19.43 cm from the sample, a voxel size of 4.2 μm was achieved, binned to a voxel size of 2.8 μm. The domain was cropped to 5.6 mm × 2.8 mm × 0.8 mm to remove edge irregularities from the razor cutting to generate the large field of view low-resolution sample. An internal scan of the sample (0.42 mm × 0.42 mm × 0.63 mm) was carried out on the 4× objective using 3201 projections and 12 s exposure time. By positioning the source 1.25 cm from the sample, and the detector 10.5 cm from the sample, a voxel size of 700 nm was reached, which constitutes the upper hardware resolution of the particular detector used in this study. All radiographs were reconstructed into 3D volumes using a cone-beam filtered back projection (FDK) algorithm using the Reconstructor software (ZEISS), as seen in Fig. 6a and c. The low-resolution imaging collection took 2 h to collect both sections of the 2-section vertical stitch covering a field of view of 4.2 mm × 7 mm, covering the entirety of the 3 mm × 6 mm sample. The high-resolution image collection took ~11 h to complete for a field of view of 0.7 mm × 0.7 mm. The protocol for obtaining a low-resolution, wide field-of-view image as well as a registered high-resolution subsample in this study requires Region of Interest (ROI), or zoom-in scanning capabilities, which are found in recent micro-CT systems designed to perform multi-scale analysis[83]. The image acquisition and processing steps are not restricted to the system in this study, and is widely applicable to any under-resolved porous structures for subsequent super resolution and multi-label segmentation.

### Super-resolution

Super-resolution of the low-resolution image in Fig. 6 was achieved using CNNs to match the quality of the high-resolution image obtained experimentally by μ-CT while maintaining the wide field of view. This methodology uses a 3D SRCNN[20] structure with a coupled pair of efficient 2D networks based on the Enhanced Deep Super-Resolution (EDSR)[73] to achieve 3D super-resolved images of large domains with minimal computational cost. Using a pair of 2D CNNs (rather than a single 3D CNN) will (i) improve training and deployment time, (ii) reduce edge effects and overlapped subdomains, and (iii) rapidly and efficiently preview and sample the SRCNN on subdomains and 2D slices. These key improvements in performance unlock the ability for large-scale super resolution of images as obtained from 3D image acquisition methods rather than being limited to small 3D domains due to GPU memory limits and CPU speed limits[20].

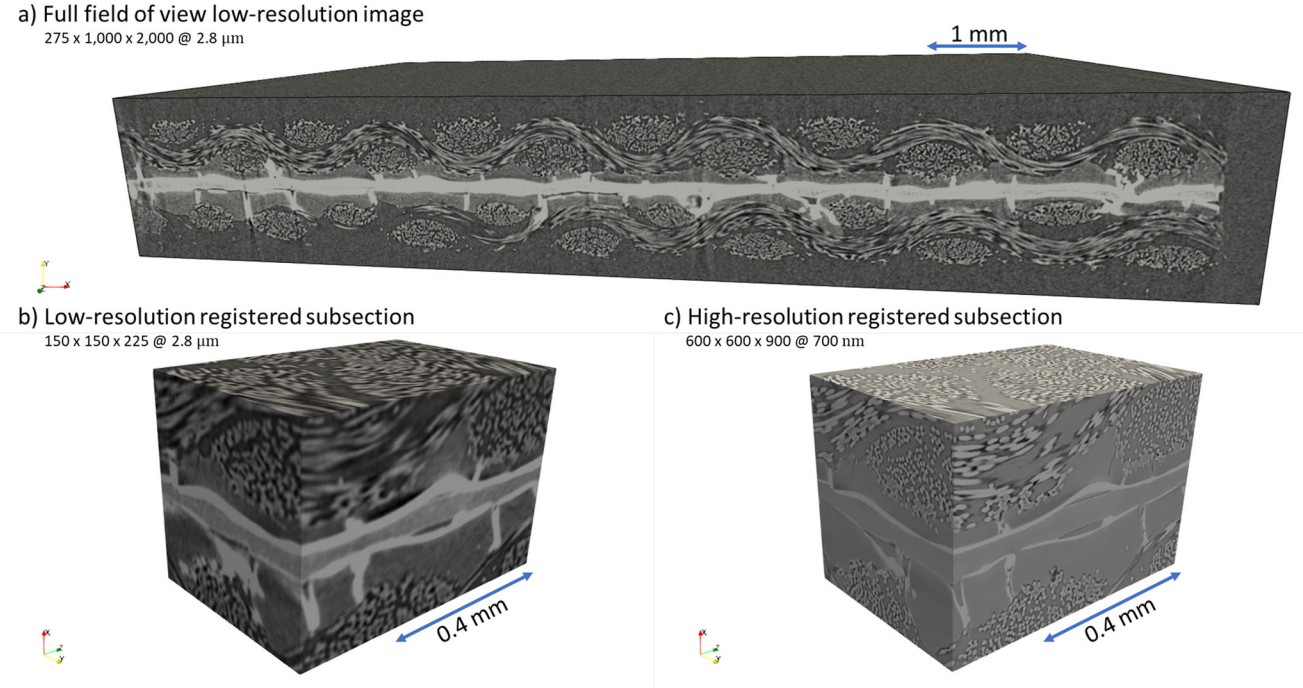

**Fig. 6 | PEMFC images as obtained by $\mu$-CT. a** Full field of view low-resolution (2.8 $\mu$m) PEMFC image, and **b** registered low-resolution and **c** high-resolution (700 nm) subdomains used for training.

This structure (DualEDSR) is a unique network developed specifically for this study, and consists of an XY super resolver $SR_{xy}$, and Z super resolver $SR_z$. During training, an input image $I_{LR}$, with shape $[N_x, N_y, N_z]$ is treated as a 2D batch of consecutive images, with the image $I_{LR}$ permuted to $I_{LRp}$ such that $shape(I_{LRp}) = [N_z, N_x, N_y, 1] \equiv [b, N_x, N_y, c]$, where $b$ is the batch size, and $c$ is the number of channels. The ground truth used to train this batch is a registered image $I_{HR}$ with shape $[N_z \times S, N_x \times S, N_y \times S, 1]$, where $S$ is the scale factor for super-resolution. In this study, this value is 4. This allows the image $I_{LRp}$ to directly be passed into $SR_{xy}$, to generate a 2D super-resolved image batch $I_{SR_{xy}}$ as $I_{SR_{xy}} = SR_{xy}(I_{LRp})$, where the shape of $I_{SR_{xy}}$ is $[b, N_x \times S, N_y \times S, c]$. The generated image $I_{SR_{xy}}$ is compared with a downsampled image $I_{HR_d}$, with its first dimension downsampled by a bicubic filter to a shape of $[N_z, N_x \times S, N_y \times S, 1]$. The loss function $L_{xy}$ computes the mean absolute error $MAE_{xy}$ between these 2 images as $MAE_{xy} = L_{xy}(I_{SR_{xy}}, I_{HR_d})$. The XY super-resolved image $I_{SR_{xy}}$ is permuted to $I_{SR_{xyp}}$ with shape $[N_x \times S, N_y \times S, b, c]$, and passed to $SR_z$ to generate a full 3D super-resolved image $I_{SR_{xyz}}$ as $I_{SR_{xyz}} = SR_z(I_{SR_{xyp}})$. The resulting image $I_{SR_{xyz}}$ has a shape of $[N_x \times S, N_y \times S, b \times S, c]$, where $c = 1$, and $b = N_z$. The input high-resolution image $I_{HR}$ is suitably permuted to $[N_x \times S, N_y \times S, N_z \times S, 1]$ as $I_{HRp}$. The loss function here $L_{xyz}$ computes the mean absolute error $MAE_{xyz}$ between these two images as $MAE_{xyz} = L_{xyz}(I_{SR_{xyz}}, I_{HRp})$. During training, both loss functions are optimised together in each iteration. The architecture is shown in Fig. 7.

DualEDSR was developed in Tensorflow 2, and training and testing were performed on an Nvidia RTX Titan GPU, and CPU testing of 3D-EDSR was performed with a 32 core AMD Ryzen Threadripper 2990WX with 128GB of RAM. The high-resolution and low-resolution images for training are split 80:20 in the z axis for training and validation. The full low-resolution image has no full size high-resolution equivalent, so it acts as an external sample. DualEDSR is trained for 500,000 iterations at a learning rate of $10^{-4}$, decaying exponentially with a half-life of 50,000 iterations. Plateau is reached well before 500,000 iterations—the extended training schedule is for investigative purposes.

3D-EDSR is also used as a comparison with DualEDSR, with the same training schedule, and its architecture is identical to the X-Y stage of DualEDSR, but with 3D layers. Differences in the performance between DualEDSR and 3D-EDSR are discussed in Supplementary Fig. 1. For comparative purposes, the same batch size of $32^3 \rightarrow 128^3$ is used for DualEDSR and 3D-EDSR. Both networks use a base filter size of 64 and residual block length of 16[40]. While unpaired CycleGANs (Cyclic Generative Adversarial Networks) could be used to train the SRCNNs without matching low and high-resolution domains[42] to train the SRCNN, such networks are more stochastic, have lower accuracy, require more computing resources and take longer to train[20]. As such, while this was not within the scope of this study, it could be potentially explored for PEMFC imaging and modelling without matching areas. Similarly, as the ultimate objective is the efficient super resolution of a large 3D image for subsequent segmentation, aspects relating to texture generation and perceptual quality with generative adversarial networks and perceptual loss functions[41] add unrequired complexity and extra computational overhead.

## Full feature segmentation

To perform multi-label segmentation on the super-resolved PEMFC dataset, a Trainable WEKA segmentation (TWS)[84] method is coupled with a CNN. This involves selecting a 2D grayscale slice from the super-resolved PEMFC and then manually clustering pixels into the five labels (pore/void, GDL, MPL, catalyst layer, membrane) as training data for Weka segmentation. The Weka segmentation generates a 2D multi-label segmented slice based on the manually clustered pixels—which are manually inspected for accuracy[47]. This then serves as a ground truth for CNN segmentation. Once the CNN is trained, the full 3D super-resolved PEMFC is segmented.

The CNN architecture used herein is U-ResNet[85,86] (Fig. 7). The training pairs include the grayscale micro-CT slice and its corresponding ground truth from Weka segmentation. It was cropped into patches with a size of $120 \times 120$ pixels. Overall, 770 patches were obtained, and split 80/20 for training and testing. The CNN model was trained for 100 epochs with an initial learning rate of $10^{-5}$ and batch

## a) Super-resolution

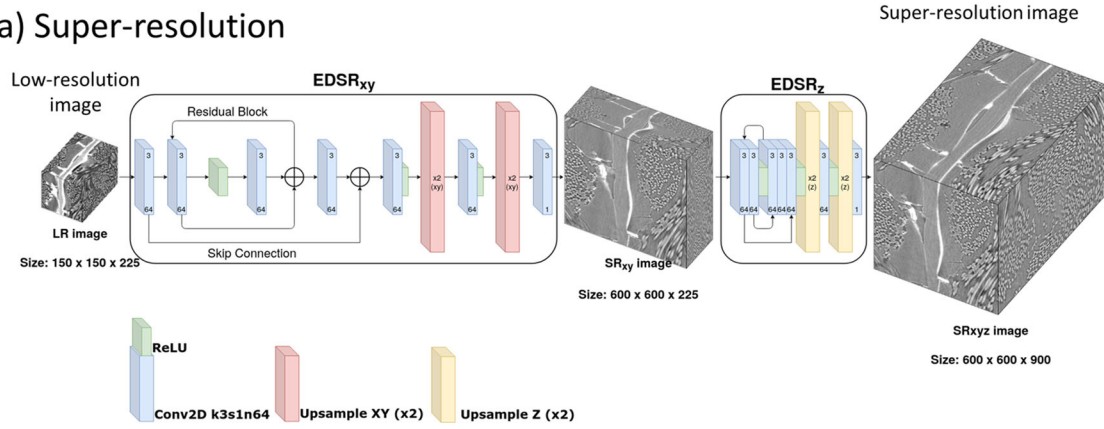

## b) Multi-label segmentation

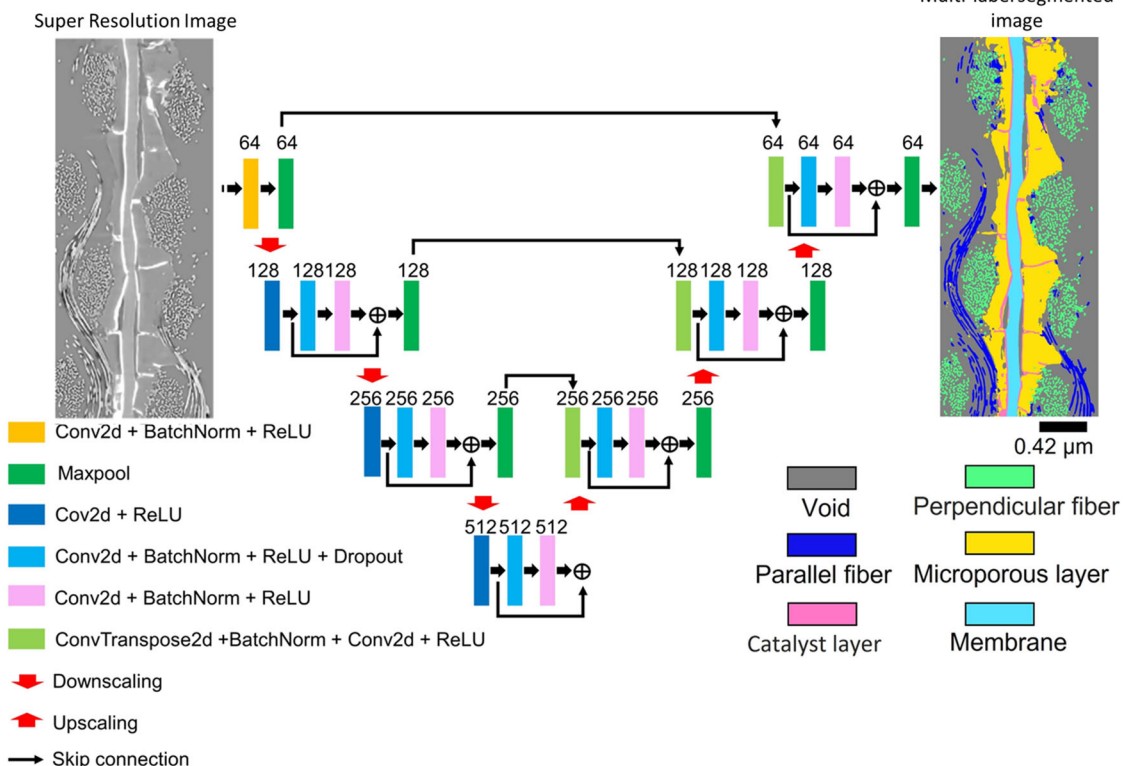

**Fig. 7 | CNN architectures for super-resolution and multi-label segmentation. a** Network architecture of DualEDSR—composed of 2 EDSR networks in series with appropriate upsampling layers. **b** Architecture of the U-ResNet, containing downsampling as image feature extraction and upsampling layer for useful feature decoding. Two fibres orientations together with the void form the GDL.

size of 16 using the Adam solver[87]. The learning rate was reduced by a factor of 0.5 each time the loss reaches a plateau for 5 epochs. The training and testing was implemented in PyTorch, using an Nvidia RTX 3090 GPU and a 16 core CPU (AMD 5950X) with 128 GB of RAM. The approximate training time per epoch was 4 s, the model was trained in 7 min for 100 epochs. After training, the other 4599 full-size (8400 × 1280) 2D slices were fed to the U-ResNet one-by-one for multi-label segmentation.

Lastly, an analogue to a membrane electrode assembly sandwiched between parallel flow fields is artificially generated by overlaying corrugated flow field plates with gas channels on both GDLs. Flow channels of 0.2 mm width and height separated by 0.2 mm lands are used as these are one of the most promising flow field configurations as well as approximately matching the half-wavelength of the GDL weave[88,89]. The gas and water interactions at the cathode are investigated using two-phase flow simulations on the upper half of this domain (Fig. 1).

### Analysis of heterogeneity

Once super-resolved and multi-label segmented, the individual layers of a PEMFC still contain heterogeneity yet to be identified and isolated. Geometric features of interest include the spatial distribution of intrusions of the catalyst layer into the MPL fractures and intrusions of the MPL into the GDL weave. As these intrusions are not segmented from the remaining materials, isolating them is non-trivial. While advanced machine learning methods such as Weka[84] can be applied, a simpler approach is proposed here by considering the material layer-by-layer hierarchy. Since the intrusions occur predominantly into the PEMFC thickness, they can be quickly isolated and quantified through projected height-maps and a semi-variogram[90].

As the PEMFC image is already oriented parallel to the width and breadth planes, no extra image rotation is required. However, if such rotation is needed, it should be performed before the segmentation step, either on the low-resolution image, or on the super-resolved image. Once segmented as per the previous section, the catalyst layer and the MPL can be masked out of the image. Once this is done, the projection of the thickness of these layers can be generated as a 2D map of the height values. This allows a basic understanding of the spatial heterogeneity in the plane of the PEMFC layers. With these 2D height maps, a variogram analysis can be performed, which calculates the semi-variance $\gamma(h)$ of all height values in an annular envelope $[\frac{h}{2}, \frac{3h}{2}]$ for values of $h$ (in voxels) up to the length of the sample. The range $h$ where $\gamma(h)$ reaches plateau (the sill of the semi-variogram) represents the length scale $h$ where the variation in height values becomes constant, representing a minimum required field of view to capture a representative sample of the underlying heterogeneity.

### Direct flow simulation

In the resolved void space of the PEMFC image, the flow of gas and water is simulated by LBM using a Multi-Relaxation Time (MRT) scheme in D3Q19 quadrature scheme for momentum transport and multi-component D3Q7 scheme for mass transport. This has been shown to be sufficient in cases of low Mach number flows for accurately representing momentum transport anisotropy for the viscous stress tensor[91–93]. The implemented multi-phase LBM has been used extensively to model flow through a complex geological materials with different flow configurations and wettabilities[94–98], demonstrating the ability to capture expected experimentally observed trends with wettability in complex geometries[99]. The microstructure of the GDL and/or MPL lends itself well to such LBM formulations for water management modelling[39,57,58,63–66]. LBM reformulates the continuum mechanics of the Navier-Stokes Equations (NSE) from underlying kinetic theory. A bulk collection of particles within a control volume has its kinetics estimated with a 19-vector velocity space $\xi_q$ and velocity distributions $f_q$. For each of the 19 entries in $\xi_q$, the velocity in the specified direction is $f_q$. Thus, an equation detailing the development of fluid transport can be constructed. The momentum transport equation at location $\vec{x}_i$ over a timestep $\delta t$ takes the form in Eq. (1) that relies on a collision operation $J$ which is model specific and outlined in detail in ref. [51]:

$$f_q(\vec{x}_i + \vec{\xi}_q \delta t, t + \delta t) = f_q(\vec{x}_i, t) + J(\vec{x}_i, t) \qquad (1)$$

The PEMFC image has its single-phase flow simulated within the pore space of the segmented image. The GDL and the open void space above it are the key regions examined. In the generation and modelling of flow in porous media, the flow paths must be adequately resolved to capture flow fields correctly. One metric used for determination of the critical image resolution is the permeability $K$ by Eq. (2):

$$K = \frac{\mu \vec{v} L}{\Delta P_x} \qquad (2)$$

where $\mu$ is the kinematic viscosity, $\vec{v}$ is the mean velocity within the bulk domain, $L$ is the length of the sample in the direction of flow, and $\Delta P$ is the pressure difference between the inlet and outlet. The single phase flow simulations were performed on the Gadi Supercomputer (National Computational Infrastructure, Australia). This implementation does not include the effects of flow in the microporous region of the MPL, which would require a Stokes-Darcy-Brinkman approach[49]. This is a target of future development and testing of an LBM implementation incorporating Darcy collision equations in the microporous region—though this is not in the scope of this study.

Multi-phase flow is solved within the pore space using Colour Gradient LBM, of which the specific implementation is detailed in

refs. [51,52]. The high-resolution of the PEMFC image permits an accurate presentation of the pore space of the GDL, the wide field of view allows the simulation of water transport and removal via the MPL fractures and GDL inter-weave holes. The multi-label segmentation allows one to assign contact angles and other material-specific properties to each layer independently. The two-phase flow simulations were performed on the Summit (Oak Ridge Leadership Computing Facility, USA) and Gadi (National Computing Infrastructure, Australia) supercomputers. While the use of particularly powerful computational resources enables direct modelling on the super resolved domain, cloud-based systems routinely support systems with 8-16 GPU, sufficient to model the full thickness of the cell at the resolution considered in this study, though at a narrower planar field of view. A realistic domain size for such cloud-based simulation using a GPU based multi-phase LBM solver would be in the approximate range of $800 \times 800 \times 600$ to $1000^3$. It is conceivable that smaller scales could still provide meaningful calculations, though larger scale simulations provide the basis to assess what system size is needed to make an adequate representation of the physical operation of the cell as well as capture any wide-scale heterogeneities that are singular or non-existent at smaller scales (such as fractures in the MPL). GPUs as a cloud computing service is expected to grow at 40% yoy[100] to 2030, which by then would reach the same order of magnitude as the relatively uncommon resources used in this study.

The multi-phase LBM in this study is implemented on a two-phase immiscible system. Flow in the gas channel and GDL is modelled, while the MPL is modelled as a hydrophobic surface that generates both water and gas at a constant, uniformly distributed gas-liquid saturation. A simple gas channel of parallel square channels of length 0.2 mm is imposed over the PEMFC image and overlapped by 30% into the GDL to emulate a compressed PEMFC. A volume render of this domain is shown in Fig. 1. This treatment thus assumes that (i) the generation of water in the catalyst layer is homogeneous, and will flow homogeneously to the surface of the MPL, thus assuming the MPL (excluding resolved fractures) is a homogeneous porous media that follows Buckley-Leverett[80] two-phase flow, and (ii) the water generated emerges fully condensed along the surface of the MPL and the gas phase contains no information regarding the moisture content. The MPL surface saturation value can be set to mimic operating conditions whereby the amount of condensed liquid water within the MPL is relatively small compared to the flow of vapour i.e. it is not being flooded, or mimic the water-air dynamics of an MPL with a different relative permeability or operating condition, such as near flooding conditions approaching a 1:1 ratio or higher. Effectively, this treatment considers water removal as the rate-limiting factor in the operation of the PEMFC. In this situation, excess water within the fuel cell creates congestion at the MPL, and strategies to efficiently remove water can enhance the overall performance of the cell[60]. This simulation thus does not directly simulate reactions in the catalyst layer, flow through the MPL, or moisture content in the gas. Furthermore, flow due to temperature gradients are captured by the imposition of a pressure gradient in lieu of a temperature gradient[64,65]. The pressure is equal to one third the trace of the stress tensor, and in situations where multiple driving forces are present (e.g. gradients in temperature, chemical potential or electrical potential), they contribute to the hydrodynamic pressure gradient as the driving force that governs momentum transport (with anisotropic stress tensor terms). Temperature can matter for other reasons, such as the local equation-of-state. Due to thermal capacity heterogeneity and phase change over the mass transport field, the thermal gradients over the compressible gas phase would be more heterogeneously distributed over the PEMFC compared to a body force. This could effect the liquid-gas interfacial tension and even influence the local wetting state. However, these effects are likely to be second order compared to the dominant behaviours, which are due to the capillary number effect. The applied

**Article**

body force rather, facilitates investigation of influence of the porous structure of the PEMFC on water management, which is the case in this study. Possible ways to incorporate these thermal-electro-chemical effects into a multi-phase LBM routine respectively are; coupling with Phreeqc[101], coupling a Poisson pressure solver and advection of saturation in the MPL to the open-pore flow in the GDL, addition of equations of state into multi-phase and multi-component LBM[55,102]. The multi-phase simulations in this study efficiently model water transport, diffusion and removal, as well as an initial starting point for future LBM modelling specific of PEMFCs.

## Data availability
The super-resolved and multi-label segmented PEMFC data generated in this study as well as the training data have been deposited in the Zenodo database under https://doi.org/10.5281/zenodo.7470938. Configuration files for the flow simulation are also available here.

## Code availability
Deep Learning code and configuration files for super resolution is available at https://github.com/yingDaWang-UNSW/Dual-EDSR and flow simulation code is available at https://github.com/OPM/LBPM. The code is also available upon request.

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

## Acknowledgements

This research used resources of the Oak Ridge Leadership Computing Facility at the Oak Ridge National Laboratory, which is supported by the Office of Science of the U.S. Department of Energy under Contract No. DE-AC05-00OR22725. This research was undertaken with the assistance of resources from the National Computational Infrastructure (NCI Australia), an NCRIS enabled capability supported by the Australian Government. C.Z. and Q.M. acknowledge the Australian Research Council for financial support (FT170100224, LP200100255, IC200100023). D.B. and P.S. acknowledge the Royal Academy of Engineering for financial support of their time (RCSRF2021/13/53 and CiET1718/59).

## Author contributions

Y.D.W., Q.M., R.T.W., F.I., C.Z., and R.T.A. conceived the study. Y.D.W. developed and deployed the deep learning algorithms for super

resolution. K.T. and Y.D.W. developed the deep learning algorithms for segmentation, and K.T. performed the segmentation and analysis. J.E.M. developed the LBM software and Y.D.W. modified and applied it to this study. R.T.W. conducted the micro-CT scans and S.T.K. provided additional detail. Y.D.W. processed the images, super resolved the domain, and performed the subsequent analysis and flow simulation. Q.M performed the manual segmentation of the low-resolution domain using Avizo. M.M.C. provided discussion and context toward PEMFC structure and modelling conditions. D.J.L.B., P.R.S., Q.M., C.Z., P.M., and R.T.A. provided supervision, resource management and review of the methodology and manuscript.

## Competing interests

The authors declare no competing interests.

## Additional information

**Peer review information** *Nature Communications* thanks Jasna Jankovic

and the other anonymous reviewer(s) for their contribution to the peer review of this work. Peer review reports are available.

