## [Peer review file · Nature Communications]

REVIEWER COMMENTS

Reviewer #1 (Remarks to the Author):

This work builds upon LBM modeling of two-phase flow in fuel cell porous media (i.e., Gas Diffusion Layer (GDL), Micro Porous Layer (MPL), and Catalyst Layer (CL)). It is certainly relevant as there is increasing interest in the topic. The authors' use of machine learning to enhance the image quality prior to segmentation is novel in this field as far as the reviewer is aware. The authors also benchmark the LBM technique using a very large, high-resolution lattice domain. The reviewer would, however, like to see the following response and/or revisions for the work to fit well into a Nature Communications publication:

1. Electrochemical models have been implemented with LBM recently, and that work should be acknowledged. Specifically, there are publications from Satjaritanun and Shimpalee among others that utilize LBAM (JES 2021 and 2020 Satjaritanun et al. DOI: 10.1149/

1945-7111/abf217 and DOI: 10.1149/2.0162001JES) and co-simulation (JES 2019 Shimpalee et al. DOI: 10.1149/2.0291911jes). Justify excluding an electrochemical model and how that may impact the authors' outcomes especially this manuscript has been focused on polymer electrolyte membrane fuel cell where electrochemical model is needed.

2. The reviewer doesn't think D3Q19 was sufficient in the complex structure in GDL, MPL, and CL. The authors need to use D3Q27. Otherwise please state the reason.

3. The reviewer does not see widespread adoption of direct modeling at these scales anytime soon. However, the imaging techniques have immediate benefits for those in direct modeling. They would be useful even at smaller scales. How do the authors want others to utilize these methods? What standards should they follow? How far-reaching is the authors' technique? Use these thought points to strengthen the results and discussion sections.

Reviewer #2 (Remarks to the Author):

In this work, the authors claim a breakthrough in achieving super resolution of a PEMFC MEAs, using Micro-CT data and deep learning to achieve super-resolution, followed by multi-label segmentation, and

direct multi-phase simulation. Using this approach, a high resolution was achieved on a large field of view, enabling a large volume simulation.

This approach seems, indeed, very useful and interesting. Can it be called a breakthrough? Possibly, if the effort to apply the deep learning is much lower than taking a number of HR micro-CT scans and stitching it together (which would achieve a similar effect). The deep learning approach is sound and novel.

Here is some feedback:

- The manuscript would be much improved if some additional clarification is added. Please see in the attached document where this is needed.
- It would be useful if the authors compare the results of the multi-phase simulation using a low-resolution data (large FOV, micro-CT data) vs. the high-resolution data (large FOV, HR obtained by deep learning). This would help to understand the value of the reported approach, how much benefit is achieved vs. how much effort (and computational time) is invested.
- Is there any comparison with the experimental data? We will never know how closely the models describe our systems, if we cannot compare to the experimental data. This is not a request to do it in this manuscript (because this would be a major work), but at least to comment or plan this work in the future.
- Some details in the methods is missing or it is not clear. This needs to be improved. Comments are in the attached document.
- Figure numbering is off, and it needs to go in the order from Fig 1 and on.
- Please see all other comments in the attached document.

Main recommendation: Major revisions needed.

Reviewer #3 (Remarks to the Author):

The paper investigates polymer electrolyte membrane fuel cell characterization by X-ray micro computed tomography. In this work the authors introduced and adapted to PEMFC technology a methodology, already proposed for the analysis of rocks, that allows to enhance image resolution. This approach is then coupled with a Lattice-Boltzmann model, demonstrating the capability to deepen the understanding of liquid water transport in PEMFC porous media and provide innovative advanced characterization. The work is well presented and clear, the topic is original and the methodology fully described. Few findings are reported from the results of the analysis, in particular: accumulation of water under the land area, roles of cracks in MPL, role of hydrophobic/hydrophilic regions in the GDL. The impact of these findings on the technology is anyway limited, because already identified in previous

works, see for example the review Weber A. et al "A Critical Review of Modeling Transport Phenomena in Polymer-Electrolyte Fuel Cells" J ECS 161 F1254 (<https://iopscience.iop.org/article/10.1149/2.0751412jes/pdf>). Anyway, the originality of the methodology provides a relevant contribution to the field, thus I suggest this paper for publication.

Few typos and points that need clarification are here reported:

- page 5 line 72: here the authors refer to anode and cathode process accordingly to the boundary conditions that they have chosen and described later (Section 2.5). I suggest to clarify this sentence that is not clear at this early point in the manuscript.
- page 10 line 188: to me, there is no justification to avoid considering liquid water on the anode side based on the fact that water is generated at the cathode. A reference from literature should be reported to support this hypothesis.
- page 11 line 191: "An air stoichiometry of 9 was set on the surface of the MPL" this sentence should be improved and clarified. Stoichiometry in technical literature indicates the non-dimensional flow rate that is fed to the channel, proportional to the consumed reaction rate. How this was set on the MPL surface is not clear.
- page 11 line: the author should indicate that the findings are associated to the specific GDL material that was used (carbon cloth). Additional information on the commercial materials that were used should be reported in the methodology.
- page 18 line 342 "The a bulk collection of particles within a control volume is has its kinetics..." seems to be a typo.
- in the manuscript, sometimes the authors refer to "ribs" and sometimes to "lands" to describe the contact region between graphite plate and GDL, I suggest to use just one of them.
- the authors do not consider temperature gradients across the porous media that have a key role in liquid water distribution and multiphase flow. Such an hypothesis should be discussed and some consideration on this limitations may be reported. The only reference to this aspect is found in page 11 line 223.

NCOMMS-22-32485 Response 1

October 2022

We thank the reviewers for their detailed comments and constructive suggestions to this manuscript. We have herein addressed all comments (in black) given to us, and **our responses and quotes from the original submission are in blue**, while our changes to the manuscript are in red.

Reviewer #1 (Remarks to the Author):**Comment #1, Reviewer #1**

This work builds upon LBM modeling of two-phase flow in fuel cell porous media (i.e., Gas Diffusion Layer (GDL), Micro Porous Layer (MPL), and Catalyst Layer (CL)). It is certainly relevant as there is increasing interest in the topic. The authors' use of machine learning to enhance the image quality prior to segmentation is novel in this field as far as the reviewer is aware. The authors also benchmark the LBM technique using a very large, high-resolution lattice domain. The reviewer would, however, like to see the following response and/or revisions for the work to fit well into a Nature Communications publication:

1. Electrochemical models have been implemented with LBM recently, and that work should be acknowledged. Specifically, there are publications from Satjaritanun and Shimpalee among others that utilize LBAM (JES 2021 and 2020 Satjaritanun et al. DOI: 10.1149/1945-7111/abf217 and DOI: 10.1149/2.0162001JES) and co-simulation (JES 2019 Shimpalee et al. DOI: 10.1149/2.0291911jes). Justify excluding an electrochemical model and how that may impact the authors' outcomes especially this manuscript has been focused on polymer electrolyte membrane fuel cell where electrochemical model is needed.

We thank the reviewer for suggesting these relevant points and have included the references suggested. A detailed discussion of this modeling aspect has been added to this revision, with comments added to the introduction. Furthermore, a discussion of the different approaches between full electrochemical modeling and water management modeling has been provided in the introduction.

The key takeaways are that the work mentioned by Shimpalee et al. over the recent years has been to combine and couple different modeling methods and different physics at different length scales, which is consistent with the multiscale nature of PEMFCs. This involves use of direct CFD in the gas channels, and LBAM/LBM in the porous layers, with reactions, phase change, and electrokinetics to model the full operation of a PEMFC. These works are acknowledged in the introduction, the discussion, and direct flow simulation Section 4.5. While indeed an electrochemical model would be needed to model the full operations of a PEMFC, this study utilises the generated domain for the purposes of water management modeling.

With these considerations in mind, commentary regarding these works have been given in the introduction.

“Furthermore, electrochemical models may be incorporated into flow modeling at various length scales using direct and agglomeration approaches with co-simulation [27, 26, 25].”

and:

“Such large simulation domains require considerable computational intensity, with even higher requirements for multiphase flow modeling of water and gas transport [35] or even electrochemical modeling of multi-component reactive transport simulations [25]. As such, due to these limits, two paradigms have emerged in modeling PEMFC operating dynamics; (i) reduced physics simulations of single or immiscible two-phase flow directly on small subsamples of GDL and MPL porous structures for water management modeling [16, 24, 22], and (ii) electrochemical simulations with co-simulation to model the operation of a PEMFC with agglomeration techniques to characterise the multi-layered, multi-scale porous structure [27, 26, 25].”

The multiphase LBM in this work focuses specifically on water management within the PEMFC and is consistent with other studies on water management dynamics within the porous structure of PEMFCs [16, 24, 22]. We expand the introduction with the following details:

“In essence, water removal dynamics in PEMFCs do not require every individual sub-process to be modeled. This is because (i) PEMFC efficiency can be limited by the timescale to remove water; and (ii) the timescale of water removal is dominated by the capillary number (true at length scales of μm to mm), viscosity and density ratio. This limiting behaviour has been observed experimentally using neutron imaging, revealing a direct correlation between liquid water accumulation and voltage losses over a short timescale [7].”

Section 2.5 detailing possible pathways to electrochemical modeling have references these works recommended by the reviewer.

“Possible ways to incorporate these features into a multi-phase LBM routine respectively are; coupling with Phreeqc [3], coupling a Poisson pressure solver and advection of saturation in the MPL to the open-pore flow in the GDL, addition of equations of state into multi-phase LBM [40, 25].”

Minor changes and extra references are added in Section 4.5 regarding electrochemical modeling in PEMFCs using LBM methods.

“Finally, the multi-phase flow model could be further enriched using electrochemical modeling considering heterogeneous current densities and heat generation [27, 26, 25]”

We also direct the reviewer to sections within the original submission that briefly discuss the limitations and future developments of water management modeling to include electrochemistry in Section 4.5 of the original file:

“Effectively, this treatment considers the water removal as the rate-limiting factor as compared to reactions that occur at the MPL. In this situation, excess water within the fuel cell creates congestion at the MPL, and strategies to efficiently remove water can enhance the overall performance of the cell [7]. This simulation thus does not directly simulate reactions in the catalyst layer, flow through the MPL, or moisture content in the gas. Possible ways to incorporate these features into a multi-phase LBM routine respectively are; coupling with Phreeqc [3], coupling a Poisson pressure solver and advection of saturation in the MPL to the open-pore flow in the GDL, and addition of equations of state into multi-phase LBM [40]. The multi-phase simulations in this study efficiently model water transport, diffusion and removal, as well as an initial starting point for future LBM modeling specific of PEMFCs.”

Comment #2, Reviewer #1

2. The reviewer doesn't think D3Q19 was sufficient in the complex structure in GDL, MPL, and CL. The authors need to use D3Q27. Otherwise please state the reason.

For low Mach number flows the critical factor in choosing the lattice structure for momentum transport able to accurately represent the anisotropy for the viscous stress tensor. On this basis, The D3Q19 model has been extensively validated for flow through porous media and it is well-established that the associated D3Q19 multi-relaxation time (MRT) collision model is able to accurately represent momentum transport in fluid systems [1]. The validity of the D3Q19 lattice structure has also been previously assessed based on asymptotic methods [13]. Other authors have elected to use the even simpler D3Q13 lattice structure, since even this simpler model can satisfy the invariance criteria needed to represent the stress tensor [31]. Our two-fluid flow lattice Boltzmann codes have been used extensively to model flow through a wide range of complex geological materials based on many different flow configurations [17, 8, 10, 9, 2]. Direct comparisons between simulations and experiments have even been performed to demonstrate that our codes are able to capture expected trends with wettability in complex geometries [18]. Similarly, representing the microstructure of the GDL and/or MPL does not introduce any unique challenges beyond what is typically encountered in digital rock physics applications and D3Q19 is extensively used in water management modeling of PEMFCs [16, 24, 22]. We also note that our codes are open source and available to the community through the Open Porous Media Project <https://github.com/OPM/LBPM> which has been added to the Data Availability Section. We believe the source of this discrepancy between D3Q19 and D3Q27 comes from the different use case of LBM and LBAM in the reviewers' previous comment on

EK modeling works by Shimpalee et al, whereby Multiphysics and agglomeration averaging is employed. This may indeed require a D3Q27 lattice structure. On the other hand, two-fluid flow lattice Boltzmann such as the one used in this study do not require this. Electrokinetic solvers have also been developed using the D3Q19 model, though constraints on the lattice weights may be different for LBMs targeted to solve the Poisson equation [40, 5, 34, 33, 32]. Extra references have been added to support the use of D3Q19 in the case of two-fluid flow simulation in complex porous media in Section 4.5:

“This has been shown to be sufficient in cases of low Mach number flows for accurately representing momentum transport anisotropy for the viscous stress tensor [1, 13, 31]. The implemented multi-phase LBM has been used extensively to model flow through a complex geological materials with different flow configurations and wettabilities [17, 8, 10, 9, 2], demonstrating the ability to capture expected experimentally observed trends with wettability in complex geometries [18]. The microstructure of the GDL and/or MPL lends itself well to such LBM formulations for water management modeling [16, 23, 12, 14, 11, 24, 22].”

Comment #3, Reviewer #1

3. The reviewer does not see widespread adoption of direct modeling at these scales anytime soon. However, the imaging techniques have immediate benefits for those in direct modeling. They would be useful even at smaller scales. How do the authors want others to utilize these methods? What standards should they follow? How far-reaching is the authors’ technique? Use these thought points to strengthen the results and discussion sections.

We thank the reviewer for these comments, we will address these questions regarding the imaging methods sequentially and provide commentary on the direct modeling capabilities. These additions are added to each relevant section of the imaging results (image acquisition+super-resolution, segmentation, and direct modeling)

How do the authors want others to utilize these methods? What standards should they follow? How far-reaching is the authors’ technique?

Firstly, the code base for this paper is fully open-source, and we have added the explicit “Data Availability” section at the end of the manuscript to reflect this (which was intended to be present in the final submission but was omitted in the original submission).

“ Deep Learning code for super resolution and segmentation is available at <https://github.com/yingDaWang-UNSW> and flow simulation code is available at <https://github.com/OPM/LBPM>. The PEMFC image is available at <https://zenodo.org/deposit/7278555> ”

In general, the hardware and software for deep learning applications is relatively mature and standardised, which is reflected in the code-base and the use of workstation-grade hardware.

Overall, in the Methods Sections 4.1 to 4.5, methodology is outlined in detail, though there are some aspects of utilisation, standards, and generalisability that could be added. As such, we have added extra recommendations for the utilisation, standards of practice, and generalisability in the results, methods, and discussion as per the paragraphs below.

Image acquisition of a high-resolution and low -resolution image should be performed in a way that allows these images to be registered in a way such that the high-resolution, low field of view image is a subset of the low-resolution, wide field of view image. The reasons for this are outlined in Section 4.2 of the original manuscript:

“While unpaired CycleGANs (Cyclic Generative Adversarial Networks) could be used to train the SRCNNs without matching low and high-resolution domains [20] to train the SRCNN, such networks are more stochastic, have lower accuracy, require more computing resources and take longer to train [36].”

This requires the image acquisition to be performed with hardware capable of region of interest imaging, and this detail is added to the manuscript in Section 4.1 on image acquisition, regarding utilisation, steps, and generalisability:

“The protocol for obtaining a low-resolution, wide field of view image as well as a registered high-resolution subsample in this study requires Region of Interest (ROI), or zoom-in scanning capabilities, which are found in recent micro-CT systems designed to perform multi-scale analysis [4]. The image acquisition and processing steps are not restricted to the system in this study, and is widely applicable to any under-resolved porous structures for subsequent super-resolution and multi-label segmentation.”

Regarding super-resolution, in Section 4.2 on super-resolution, the methodology describes the usage, standards, and generalisability already in the original manuscript:

“This methodology uses a 3D SRCNN structure with a coupled pair of efficient 2D networks to achieve 3D super-resolved images of large domains with minimal computational cost. Using a pair of 2D CNNs (rather than a single 3D CNN) will (i) improve training and deployment time, (ii) reduce edge effects and overlapped subdomains, and (iii) rapidly and efficiently preview and sample the SRCNN on subdomains and 2D slices.”

To expand further on this, extra detail is added to Section 4.2:

“...reduce edge effects and overlapped subdomains, and (iii) rapidly and efficiently preview and sample the SRCNN on subdomains and 2D slices. These key improvements in performance unlock the ability for large-scale super-resolution of images as obtained from 3D image acquisition methods.”

In Section 2.1 on super-resolution, we add extra discussion on both image acquisition and super-resolution to emphasise the benefits and generalisable, far-reaching applicability of the method:

“In terms of time efficiency, training Dual-EDSR took under 12 hours to reach plateau on a single RTX Titan GPU (see Supplementary Fig. 2), and generation of the super-resolved $1,100 \times 4,000 \times 8,000$ voxels @ 700 nm resolution image took under 1 hour of GPU time. In comparison, if one were to attempt to generate this high-resolution, wide field of view image by zoom-in and stitching regions of interest, the single small high-resolution training block (Fig. 6) took approximately 11 hours to acquire, for an FOV of $600 \times 600 \times 900$ voxels @ 700 nm resolution. To collect high-resolution data across the entire sample would require approximately and at minimum (assuming no overlap of the data sets) 108 such high-resolution blocks, totalling 1188 hours of data collection (7.5 months at 8 hours a day, 5 days a week), not including data set stitching or allowing for needed overlaps to ensure proper alignment of the data sets. Furthermore, once Dual-EDSR is trained on a specific type of material and imaging condition, the procedure can be repeated. Training a new Dual-EDSR for other samples and imaging conditions is similarly straight-forward by generating new low-resolution and high-resolution images as described in the Methods Section 4.1.”

Regarding segmentation, in Section 4.3 on multi-label segmentation, similarly, the methodology and the usage, standards, and generalisability are noted as per the first 2 paragraphs of the section, and this detailed methodology is referred to in Section 2.2 of the original manuscript:

“Full feature PEMFC segmentation was performed via the CNN described in Section 4.3.”

In order to improve readability, in Section 2.2 on multi-label segmentation, we add the following extra statements:

“Full feature PEMFC segmentation was performed via the workflow and methodology described in Section 4.3 involving the generation of ground truth using machine learning and the training of a CNN to segment the entire domain, which is an established and generalizable segmentation methodology [29, 30].”

Overall, as indicated in each section (imaging, super resolution, segmentation), the hardware and software is readily available (benchtop micro-CT scanners, GPU workstations, deep learning code, flow simulation code) to be utilised with the standards this study suggests, which makes the techniques

potentially very far-reaching.

To this, we have added some extra statements regarding usage, see changes below. with regards to the hardware limitations for large-scale direct modeling, we address these below after the changes listed.

Regarding widespread adoption of direct modeling at these scales:

Alongside a discussion of the utilisation, standards, and generalisability, we will also address comments regarding the practicalities and adoption of direct modeling at these scales.

The reviewer is correct that direct modeling at this scale is accessible only using particularly powerful computational resources. However, cloud-based systems routinely support systems with 8-16 GPU which are sufficient to model the full thickness of the GDL even at the resolution considered here. LBPM [18] is already being used to perform digital rock physics simulations using commercial cloud resources. These rely on multiphase flow through complex 3D image data. Image sizes as large as $1,000^3$ are accessible using typical GPU instances that are available through AWS and Microsoft Azure. While it is indeed the case that the full-scale simulations considered in this work are too large for anything but the biggest supercomputers, we expect that meaningful simulations can be performed using commercial cloud resources for PEM fuel cells within the next five years. As such, extra discussion on multi-phase flow limitations, utilisation, standards, and generalisability has been added to the manuscript in Section 4.5:

“The two-phase flow simulations were performed on the Summit (Oak Ridge Leadership Computing Facility, USA) and Gadi (National Computing Infrastructure, Australia) supercomputers. While the use of particularly powerful computational resources enables direct modeling on the super-resolved domain, cloud-based systems routinely support systems with 8-16 GPU, sufficient to model the full thickness of the cell at the resolution considered in this study, though at a narrower planar field of view. A realistic domain size for such cloud-based simulation using a GPU based multi-phase LBM solver would be in the approximate range of $800 \times 800 \times 600$ to $1,000^3$. It is conceivable that smaller scales could still provide meaningful calculations, though larger scale simulations provide the basis to assess what system size is needed to make an adequate representation of the physical operation of the cell as well as capture any wide-scale heterogeneities that are singular or non-existent at smaller scales (such as fractures in the MPL). GPUs as a cloud computing service is expected to grow at 40% yoy [21] to 2030, which by then would reach the same order of magnitude as the relatively uncommon resources used in this study.”

On the topic of generalisability to other 3D imaging, these imaging methods can be utilized across the fuel cell imaging community and beyond to enable higher resolution imaging of larger fields of view than were previously practical. This is useful for sample inspection applications and should also find utility in quantitative image analysis and modeling applications such as the work demonstrated here. Other applications where pore network resolution and mapping detail is essential should benefit from using this method. Examples include battery electrode analysis, gas and fluid flow applications such as membrane and filter analysis, geoscience, and catalysis. Similarly, particle quantification applications in fields such as additive manufacturing, mining, and soil research could benefit from this technique. Defect inspection and failure analysis applications could also benefit from this capability as the added resolution across large fields of view could allow for higher defect quantification levels and failure observations. Additional discussion of the extent of applicability of the techniques used in this study in Section 4.5:

“As such, through sensitivity analysis of large-scale PEMFC domains, an improved understanding of the porous structure influences on gas-water flow in PEMFCs will lead to more effective designs. These beyond-hardware imaging and modeling methods extend beyond fuel cell imaging to enable higher resolution imaging of larger fields of view than previously practical. Applications span simple sample inspection applications to quantitative image analysis and modeling applications such as this study where spatial detail is essential.”

Reviewer #2 (Remarks to the Author):

In this work, the authors claim a breakthrough in achieving super-resolution of a PEMFC MEAs, using Micro-CT data and deep learning to achieve super-resolution, followed by multi-label segmentation, and direct multi-phase simulation. Using this approach, a high-resolution was achieved on a large field of view, enabling a large volume simulation.

We thank the reviewer for their comments, we will address them below.

Comment #1, Reviewer #2

This approach seems, indeed, very useful and interesting. Can it be called a breakthrough? Possibly, if the effort to apply the deep learning is much lower than taking a number of HR micro-CT scans and stitching it together (which would achieve a similar effect). The deep learning approach is sound and novel.

This is an excellent point, which has been emphasized in the manuscript following the reviewer's comment, with additional information added into the manuscript in Section 2.1:

“In terms of time efficiency, training Dual-EDSR took under 12 hours to reach a plateau on a single RTX Titan GPU (see Supplementary Fig. 2), and generation of the super-resolved $1,100 \times 4,000 \times 8,000$ voxels @ 700 nm resolution image took under 1 hour of GPU time. In comparison, if one were to attempt to generate this high-resolution, wide field of view image by zoom-in and stitching regions of interest, the single small high-resolution training block 6) took approximately 11 hours to acquire, for an FOV of $600 \times 600 \times 900$ voxels @ 700 nm resolution. To collect high-resolution data across the entire sample would require approximately and at minimum (assuming no overlap of the data sets) 108 such high-resolution blocks, totalling 1188 hours of data collection (7.5 months at 8 hours a day, 5 days a week), not including data set stitching or allowing for needed overlaps to ensure proper alignment of the data sets. This is an image acquisition time reduction of at least 1 order of magnitude in the most ideal case. Furthermore, once Dual-EDSR is trained on a specific type of material and imaging condition, the procedure can be repeated. Training a new Dual-EDSR for other samples and imaging conditions is similarly straight-forward.”

Comment #2, Reviewer #2

Here is some feedback: - The manuscript would be much improved if some additional clarification is added. Please see in the attached document where this is needed.

We thank the reviewer for these comments, we will address these as part of the final comment below

Comment #3, Reviewer #2

- It would be useful if the authors compare the results of the multi-phase simulation using a low-resolution data (large FOV, micro-CT data) vs. the high-resolution data (large FOV, HR obtained by deep learning). This would help to understand the value of the reported approach, how much benefit is achieved vs. how much effort (and computational time) is invested.

We have made similar such analysis of the simulation accuracy in the Section 2.4 Permeability and Velocity Field Heterogeneity of the original manuscript:

The key finding in this section regarding the image quality is noted as: “the pore space increasingly closes off near pore space constrictions and fiber contact points, causing inaccurate no-flow regions in

what would otherwise be open flow paths”.

And this is quantified by the excerpt: “computed permeability over the downsampling factors, with zoomed-in areas of the pore-space detail for each down-sampling level. While the super-resolved PEMFC retains a reasonable level of detail when downsampled by a factor of 2 (1.4 μm resolution), further down-sampling reduces the permeability from $5.9 \cdot 10^{-8} \text{ m}^2$ to $1.45 \cdot 10^{-11} \text{ m}^2$, with a percolation threshold between 1.4 μm and 2.1 μm . Additional visual representations of the velocity can be found in Supplementary Fig. 10 and Fig. 11.”

While we could reinforce this aspect with a single-phase or multi-phase simulation comparison with the low-resolution data and the high-resolution data directly (rather than synthetically downsampling the high-resolution data), the simulation comparison is effectively unnecessary. This is because the segmentation of the low-resolution data without the deep learning approach is essentially impossible, especially in regard to identifying layers and preserving image sharpness.

We add this segmentation comparison in this revision, in supplementary Figure 6, which is referred to in sections 2.2 to 2.5 to reinforce that it is only through the enhancement of the image through the deep learning workflow, that accurate physical measurements of heterogeneity and flow simulations are possible.

We have added a new figure in supplementary figure 6:

Alongside this figure, detailed discussion is given in Section 2.2, as well as within the flow simulation sections 2.4 and 2.5 in reference to this segmentation comparison to show that flow simulation would be highly inaccurate without the super-resolution as well as the multi-label CNN segmentation. Both steps are required to accurately detail the spatial structure of the PEMFC.

Added paragraph to section 2.2:

“Furthermore, to emphasise the importance of the deep learned super-resolution and multi-label segmentation, attempts were made to segment the low-resolution image using manual segmentation (Avizo software, without deep learning) and with multi-label CNN segmentation of the low-resolution dataset. In the case of manual segmentation on the low-resolution image, the MPL was indistinguishable from the GDL and the layer failed to be segmented. In both cases, the thicknesses of the catalyst layer and GDL fibers was oversegmented by a factor of 3 or more due to image blur and lack of spatial features, resulting in excessive contact area between GDL fibers and over-estimation of catalyst deposition thickness. A visual comparison between the segmentation of the low-resolution image using (i) manual segmentation with Avizo software and (ii) multi-label CNN segmentation as outlined above is given with a further comparison with (iii) the super-resolved multi-label CNN segmentation can be found in Supplementary Fig. 6). These excessive physical inaccuracies in the pore structure preclude the possibility of accurate flow simulations on the low-resolution domain, with or without deep learned segmentation.”

Added in Section 2.4:

In the final paragraph of Section 2.3, attempts to segment the low-resolution domain already showed highly inaccurate pore structures for flow simulation. The influence of the image resolution on single-phase flow in porous media is further revealed by downsampling the segmented super-resolved image.

Added in Section 2.5:

Section 2.2 and 2.4 show that segmenting the low-resolution domain or downsampling the super-resolved domain will result in geometries unsuitable for flow simulation. As such, the full super-resolved, multi-label segmented domain is used.

Figure 1: Supplementary Figure 6: Comparison with selected zoomed in subsections of a) 2D cross section of manual segmentation using Avizo software (Fisher Scientific) on the low-resolution μ -CT image for comparison with (b) multi-label CNN segmentation on the low-resolution μ -CT image and (c) multi-label segmented and super-resolved image. The manual segmentation failed to identify the MPL from the GDL, and both segmentations on the low-resolution image suffer significantly from blur and diffuse boundaries, resulting in oversized catalyst thickness and inaccurate MPL and GDL fiber geometries.

Comment #4, Reviewer #2

- Is there any comparison with the experimental data? We will never know how closely the models describe our systems, if we cannot compare to the experimental data. This is not a request to do it in this manuscript (because this would be a major work), but at least to comment or plan this work in the future.

This is an important further step, and thus we have added a brief discussion of this as follows below, in Section 2.5:

“...water flooding with extensive water retention in steady-state conditions. These complex design considerations between water management and PEMFC porous structure can also be probed with low resolution operando experiments such as 2D neutron imaging and X-ray radiography. These would supplement the numerical modeling of the super resolved images and could be used to tune and validate water management modeling for sensitivity studies.”

Comment #5, Reviewer #2

- Some details in the methods is missing or it is not clear. This needs to be improved. Comments are in the attached document.

These have been addressed as part of the final comment below

Comment #6, Reviewer #2

- Figure numbering is off, and it needs to go in the order from Fig 1 and on.

The figures are arranged following the journal structure requesting methods at the end of the manuscript. This has been fixed in revision as references to Fig 6 in Section 2.1 are changed to refer to Section 4.2:

“The super-resolution algorithm (DualEDSR) is trained on the high- and low-resolution registered images as per Section 4.2.”

Comment #7, Reviewer #2

- Please see all other comments in the attached document.

Comment #7a, Reviewer #2

Comment: maybe add “nanoporous here”

Response: This has been added

Changes: “nanoporous electrocatalyst”

Comment #7b, Reviewer #2

Comment: Possibly should be deleted, since “microporous” term is already there.

Response: The microporous layer is commonly accepted name for the layer between the catalyst and GDL, but is intrinsically nano-porous in its pore-size distribution. We have changed this to conform with the naming convention.

Changes: “covered by a micro-porous layer (MPL)”

Comment #7c, Reviewer #2

Comment: Should keep consistent (use “-” or not)

Response: Yes, we have added hyphens to all cases.

Changes: Hyphens added

Comment #7d, Reviewer #2

Comment: It is not entirely clear what authors mean by “dual porosity” in this sentence. Large perforations and cracks? MPL or GDL porosity? Please clarify by adding something like “dual porosity due to”

Response: We have clarified that we refer to “flow between MPL/GDL pores and perforations and cracks”

Changes: This “dual-porosity” from flow between MPL/GDL pores and perforations and cracks is well-known for multi-phase flow through heterogeneous and fractured porous media “

Comment #7e, Reviewer #2

Comment (Combined): The authors should explain what a “digital twin” is. E.g., from Wikipedia: “A digital twin is a real-time virtual representation of a real-world physical system or process that serves as the indistinguishable digital counterpart of it for practical purposes, such as system simulation, integration, testing, monitoring, and maintenance.” Or “A digital twin is a virtual representation of an object or system that spans its lifecycle, is updated from real-time data, and uses simulation, machine learning and reasoning to help decision-making.” The authors should also differentiate between typical simulation and modeling and digital twin. Again, what is making this approach a “digital twin” and not a typical simulation? Once successfully segmented (using image processing or manual segmentation, or machine learning), which is often done in PEMFCs world (see work of Erick Kjeang, Iryna Zenyuk, Aimey Bazlylak, the simulation of the flow, etc. is routinely performed.

Response: The reviewer is correct in this definition of digital twin. This was due to confusion by the authors regarding the difference between “image processing and simulation” with “digital twin”. This has been fixed throughout the manuscript to reflect this study.

Changes: All references to digital twin in the manuscript changed to “image” or “domain”.

Comment #7f, Reviewer #2

Comment: What is the voxel size here?

Response: The low-resolution image is 2.8 microns.

Changes: “image of $275 \times 1,000 \times 2,000$ voxels at $2.8 \mu\text{m}$ ”

Comment #7g, Reviewer #2

Comment: I think this method requires a description and details. There are no details on how this approach was done in the main paper or in the supplemental materials.

Response: The super resolution method is described in full detail, including architecture, implementation, parameters, and hardware in Section 4.2. In Section 2.2, we give an additional brief description.

Changes: Added extra description and details in Section 2.2:

“This involves firstly imaging the whole domain at a low-resolution, and then imaging a small sub-domain at high-resolution with a region of interest scan [6]. The corresponding sub-domain within the low-resolution image is used with the high-resolution image to train DualEDSR to generate super-resolved images from other unseen low-resolution images. DualEDSR is outlined in full detail in Section 4.2, and comprises a pair of 2D EDSR networks [15] trained in tandem to efficiently super-resolve the X-Y and Z directions, facilitating the practical super-resolution of large-scale images.

Comment #7h, Reviewer #2

Comment: The voxel size is still the same as the micro-CT resolution. This is a larger domain but I am not sure if we can call it super-resolved. Later in the text it is better explained, but please explain a bit what you mean by super-resolved.

Response: We have clarified this.

Changes: “architecture to $1,100 \times 4,000 \times 8,000$ voxels at 700 nm, combining the upper limits of both resolution and field of view”

Comment #7i, Reviewer #2

Comment: References to this algorithm should be provided.

Response: Dual-EDSR is a unique network developed specifically for this study – it is novel and unpublished to date. Furthermore, its structure and inspiration and other pertinent details are outlined in Sections 2.2 and 4.2, which has been referred to in the same paragraphs.

Changes: Added extra information in Section 2.1:

“The super-resolution algorithm (DualEDSR) developed specifically for this study to handle large 3D images efficiently,”

Added extra information in Section 4.2:

“This structure (DualEDSR) is a unique network developed specifically for this study, and consists of an XY super resolver ”

Comment #7j, Reviewer #2

Comment: Were both HR and LR images (3D data sets) obtained by Micro-CT? Please clarify or add a Methods section.

Response: Yes, and this information is outlined in the method section 4.1. It is located at the end of the manuscript as per the nature communications style guide. We have added a reference to the Section to reduce confusion.

Changes: Fig 6 changed to refer to Section 4.2

Comment #7k, Reviewer #2

Comment: The Figures in the text should be in order from Fig. 1 , Fig. 2, etc...

Response: We have fixed this in the previous comments.

Changes: Fig 6 changed to refer to Section 4.2

Comment #7l, Reviewer #2

Comment: Please clarify if high resolution images were obtained by HR micro CT, while super-resolution by the algorithm? This is a bit confusing.

Response: This is something we briefly clarify here in revision.

Changes: “This involves firstly imaging the whole domain at a low-resolution, and then imaging a small sub-domain at high-resolution with a region of interest scan. The corresponding sub-domain within the low-resolution image is used with the high-resolution image to train DualEDSR to generate super-resolved images from other unseen low-resolution images.”

Comment #7m, Reviewer #2

Comment: Figure order...

Response: This has been fixed.

Changes: “The low-resolution image of the physical PEMFC sample acquired by micro-CT (details in Section 4.1)”

Comment #7n, Reviewer #2

Comment: Can any methodology be provided for this approach?

Response: We have added further in-section detail as per our response to comment 7g.

Comment #7o, Reviewer #2

Comment: It is not clear if the authors actually did HR micro CT (with 700 nm resolution) and used those images to compare the accuracy of the super-resolved data. Please clarify.

Response: Yes this comparison has been made using a registered unseen subsample. We outline this in the Section 2.1:

Changes: “The corresponding sub-domain within the low-resolution image is used with the high-resolution image to train DualEDSR to generate super-resolved images from other unseen low-resolution images. The overall validation accuracy as measured by the Peak Signal to Noise Ratio on a small unseen section of this sub-domain is 31 dB which is less than 0.1% mean squared error.” And “A visual comparison of the low-, high- and super-resolution images from the validation set can be found in Supplementary Fig. 2 alongside detailed performance comparison with 3D-EDSR [15]”

Comment #7p, Reviewer #2

Comment: space

Response: Removed.

Comment #7q, Reviewer #2

Comment: This approach does seem quite useful to improve the resolution, while keeping a large FOV. Has the accuracy been confirmed by comparison with the HR micro-CT (e.g. with 700 nm resolution) of a smaller (same) area?

Response: Yes, see our response to the previous comment.

Changes: “The corresponding sub-domain within the low-resolution image is used with the high-resolution image to train DualEDSR to generate super-resolved images from other unseen low-resolution images. The overall validation accuracy as measured by the Peak Signal to Noise Ratio on a small unseen section of this sub-domain is 31 dB which is less than 0.1% mean squared error.” And “A visual comparison of the low-, high- and super-resolution images from the validation set can be found in Supplementary Fig. 2 alongside detailed performance comparison with 3D-EDSR [15]”

Comment #7r, Reviewer #2

Comment: Change to “may penetrate”, as this is not always the case.

Response: We have changed this.

Changes: “may penetrate”

Comment #7s, Reviewer #2

Comment: Were these 3D data sets obtained by micro-CT? Please clarify.

Response: These were obtained from the super-resolution and multi-label segmentation of the low-resolution image.

Changes: “catalyst layer and MPL from the full-size super-resolved and multi-label segmented image obtained prior and depicted in Figure (d-e)”

Comment #7t, Reviewer #2

Comment: 3D data set.

Response: We have clarified this.

Changes: “probe the required computational resources for such a simulation on this large 3D dataset of a super-resolved, multi-label segmented PEMFC image”

Comment #7u, Reviewer #2

Comment: Was this simulation performed on both original 3D data sets (low resolution and super-resolved)? Please clarify.

Response: This simulation was performed by taking the super-resolved image and downsampling the domain. This was necessary as it was impossible to accurately segment the low resolution image using conventional methods. We have added this comparison between the segmentation outcomes of the low resolution image compared to the super resolved image in Supplementary Figure 6 as a part of our response to a previous comment on performing simulation on the low-resolution image..

Comment #7v, Reviewer #2

Comment: Please clarify that these images were obtained by micro-CT and not by the algorithm.

Response: These images are obtained from processing the super-resolved multi-label segmented image, which originates from the original low-resolution wide field of view micro-CT image.

Changes: “From the super-resolved image; (a) 2D projections...”

Comment #7w, Reviewer #2

Comment: Please clarify what you mean by this.

Response: We clarify in the changes below.

Changes: “the pore space increasingly closes off near pore space constrictions and fiber contact points, causing inaccurate no-flow regions in what would otherwise be open flow paths”

Comment #7x, Reviewer #2

Comment: How was water and gas flow through unresolved nano-pores addressed? Where was the boundary set? At the MPL-cathode interface? How was cathode taken into account?

Response: This consideration is outlined in full detail in methods Section 4.5, which is referred to in the previous sentence. To avoid confusion, we have moved this reference to the second sentence.

Changes: “The generation and transport of water in the large-scale PEMFC is modeled using direct multi-phase flow simulation. The super-resolution of the PEMFC image provides a representation of the pore space of the GDL while the wide field of view allows to simulate water transport and removal through the MPL fractures and GDL inter-weave holes as outlined in Section 4.5.”

Comment #7y, Reviewer #2

Comment: Ignore my comment later on compression.

Response: Understood.

Comment #7z, Reviewer #2

Comment: This is OK if only MPL and GDL are considered. Still, presence of PTFE should probably be accounted for.

Response: Yes, we note that this sample does not contain PTFE, and note how this would affect the wetting distribution.

Changes: “as this sample does not contain PTFE, which would result in more complex mixed wetting surface distributions”

Comment #7za, Reviewer #2

Comment: Ideally, in one of the next studies, these results should be verified by operando imaging of water in the same cell.

Response: Yes, we do add a discussion of experimental future work as suggested in a previous comment, but we also add a brief statement here in the discussion as suggested by this comment.

Changes: “Applications span simple sample inspection applications to quantitative image analysis to direct modeling and/or future in-operando experiments leveraging the image enhancement aspects of this study, where spatial detail is essential.”

Comment #7zb, Reviewer #2

Comment: Please specify that all these were obtained experimentally by micro-CT.

Response: We have made this specification.

Changes: “As obtained by μ -CT; a) full field of view...”

Comment #7zc, Reviewer #2

Comment: Add “obtained experimentally by micro-CT”

Response: This has been added.

Changes: “the high-resolution image obtained experimentally by μ -CT while maintaining...”

Comment #7zd, Reviewer #2

Comment: References needed

Response: Added reference to SRCNNs.

Changes: Added reference to SRCNNs.

Comment #7ze, Reviewer #2

Comment: It is hard to distinguish MPL from pores (based on the images above). Was this true?

Response: This was not the case, as the MPL greyscale value was slightly higher than the pore space. This is subtle, but is visible and was picked up by both manual segmentation and deep learning segmentation of the super resolved image.

Comment #7zf, Reviewer #2

Comment: How was accuracy ensured? Was the result compared to something? Please clarify.

Response: As this initial step in training segmentation requires a ground truth segmentation to exist, the accuracy was controlled by manual inspection, as there does not exist any prior segmentation information to compare against. Later in the training of the segmentation CNN, accuracy is ensured by comparing the deep learned segmentation with this manually generated ground truth. We have added a reference to this methodology as applied to PEMFCs.

Changes: Added a reference to PEMFC segmentation by deep learning

Comment #7zg, Reviewer #2

Comment: How was the compression of the FFP accounted for? The imaging was done on a non-compressed MEA, I suppose. Situation is different when the MEA is compressed.

Response: As requested by the reviewer, we will ignore this comment – as it has been addressed in the original manuscript already.

Reviewer #3 (Remarks to the Author):

Comment #1, Reviewer #3

The paper investigates polymer electrolyte membrane fuel cell characterization by X-ray micro computed tomography. In this work the authors introduced and adapted to PEMFC technology a methodology, already proposed for the analysis of rocks, that allows to enhance image resolution. This approach is then coupled with a Lattice-Boltzmann model, demonstrating the capability to deepen the understanding of liquid water transport in PEMFC porous media and provide innovative advanced characterization. The work is well presented and clear, the topic is original and the methodology fully described. Few findings are reported from the results of the analysis, in particular: accumulation of water under the land area, roles of cracks in MPL, role of hydrophobic/hydrophilic regions in the GDL. The impact of these findings on the technology is anyway limited, because already identified in previous works, see for

example the review Weber A. et al “A Critical Review of Modeling Transport Phenomena in Polymer-Electrolyte Fuel Cells” J ECS 161 F1254 (<https://iopscience.iop.org/article/10.1149/2.0751412jes/pdf>). Anyway, the originality of the methodology provides a relevant contribution to the field, thus I suggests this paper for publication.

We thank the reviewer for this suggestion. As noted above, the findings and analysis of the generated super resolved, multi-label segmented PEMFC are known phenomena, while the approach used is highly innovative with a broad range of applications. References have been added to the final paragraph of introduction, now reads:

“...alignment and misalignment of gas channels over the GDL [38]”

Comment #2, Reviewer #3

Few typos and points that need clarification are here reported: - page 5 line 72: here the authors refer to anode and cathode process accordingly to the boundary conditions that they have chosen and described later (Section 2.5). I suggest to clarify this sentence that is not clear at this early point in the manuscript.

We thank the reviewer for raising this point. The section has been modified to be a general discussion of assumptions and limitations in electrochemical modeling vs water management modeling. In conjunction with comment #1 from reviewer 1 regarding the additional discussion of electrochemical modeling, this section of the introduction has been altered and clarified to discuss in more detail the assumptions in modeling:

“In the former case of water management modeling, the highly intensive direct simulation on the voxels of segmented micro-CT images involves a number of simplifying assumptions. For example, since liquid water is generated at the cathode and thus primarily flows from the cathode catalyst layer to the cathode gas channels through the MPL and GDL, a simplifying assumption can be made to limit multi-phase flow modeling to the cathode side only [19]. The computational resources required to capture water generation and transport dynamics at the cathode are several orders of magnitude higher than single-phase simulations at the anode which would reach steady state conditions quickly [37].”

Comment #2, Reviewer #3

- page 10 line 188: to me, there is no justification to avoid considering liquid water on the anode side based on the fact that water is generated at the cathode. A reference from literature should be reported to support this hypothesis.

Similarly, to the previous comment and comment #1 from reviewer 1 regarding the use of electrochemical modeling vs water management modeling, the assumptions made in this study are based on a few factors, which are supported by literature and referred to in the study.

Extra information has also been added to the paper to further reinforce the modeling based on water at the cathode. This reference [19] shows that limited back diffusion occurs when using weaved gas diffusion layer and relatively uniform MPLs. While of course a degree of water will be present at the anode, it is not the limiting factor in water management compared to the cathode, which is why only cathode water management features were considered here. As such, this has been added to Section 2.5:

“As the water is generated at the cathode, this model solely considers the upper half of the cell. This is assumed as limited back diffusion occurs when using a weaved gas diffusion layer and relatively uniform MPLs [19]. While some water will be present at the anode, it is negligible in terms of management of its removal, which is rate-limited by the cathode”

Comment #3, Reviewer #3

- page 11 line 191: “An air stoichiometry of 9 was set on the surface of the MPL” this sentence should be improved and clarified. Stoichiometry in technical literature indicates the non-dimensional flow rate that is fed to the channel, proportional to the consumed reaction rate. How this was set on the MPL surface is not clear.

The details on how the MPL surface is assigned is found in the Methods Section 4.5:

“the MPL is modeled as a hydrophobic surface that generates both water and gas at a constant, uniformly distributed rate and stoichiometry”

and:

“This treatment thus assumes that (i) the generation of water in the catalyst layer is homogeneous, and will flow homogeneously to the surface of the MPL, thus assuming the MPL is a homogeneous porous media that follows Buckley-Leverett [28] two-phase flow, and (ii) the water generated emerges fully condensed along the surface of the MPL”

To improve readability, extra detail as been added to Section 2.5:

“An air stoichiometry of 9 was set on the surface of the MPL to reproduce a differential cell configuration [39], as described in Section 4.5, whereby the gas/liquid fraction boundary condition is set to 9 on the MPL surface”

and in Section 4.5:

“Flow in the gas channel and GDL is modeled, while the MPL is modeled as a hydrophobic surface that generates both water and gas at a constant, uniformly distributed rate and stoichiometry (gas liquid fraction)”

Comment #4, Reviewer #3

- page 11 line: the author should indicate that the findings are associated to the specific GDL material that was used (carbon cloth). Additional information on the commercial materials that were used should be reported in the methodology.

In terms of additional information, extra details have been added to the first paragraph of Section 3.1:

It now reads: A commercially prepared 25 cm² membrane electrode assembly composed of two woven carbon fiber cloth GDLs with hydrophobic MPLs (410 μm thick, W1S1011, CeTech), and catalyst layers (0.5 mg_{Pt} cm⁻², 60% Pt/C on Vulcan) sandwiching a perfluorosulfonic acid proton exchange membrane (50.8 μm thick, Nafion NR-212, Dupont) was used for this work (FuelCellStore, US).

Comment #5, Reviewer #3

- page 18 line 342 “The a bulk collection of particles within a control volume is has its kinetics...” seems to be a typo.

This has been fixed to read:

“A bulk collection of particles within a control volume has its kinetics...”

Comment #6, Reviewer #3

- in the manuscript, sometimes the authors refer to “ribs” and sometimes to “lands” to describe the contact region between graphite plate and GDL, I suggest to use just one of them.

“ribs” has been replaced with “lands” for consistency throughout and with figure 1.

Comment #7, Reviewer #3

- the authors do not consider temperature gradients across the porous media that have a key role in liquid water distribution and multiphase flow. Such an hypothesis should be discussed and some consideration on this limitations may be reported. The only reference to this aspect is found in page 11 line 223.

The treatment of flow fields due to temperature gradients is discussed in more detail as follows. The key point is outlined in Section 4.5:

“Furthermore, flow due to temperature gradients are captured by the imposition of a pressure gradient in lieu of a temperature gradient [12, 14]. The pressure is equal to one third the trace of the stress tensor, and in situations where multiple driving forces are present (e.g. gradients in temperature, chemical potential or electrical potential), they contribute to the hydrodynamic pressure gradient as the driving force that governs momentum transport (with anisotropic stress tensor terms). Temperature can matter for other reasons, such as the local equation-of-state. Due to thermal capacity heterogeneity and phase change over the mass transport field, the thermal gradients over the compressible gas phase would be more heterogeneously distributed over the PEMFC compared to a body force. This could effect the liquid-gas interfacial tension and even influence the local wetting state. However, these effects are likely to be second order compared to the dominant behaviors, which are due to the capillary number effect. The applied body force rather, facilitates investigation of influence of the porous structure of the PEMFC on water management, which is the case in this study. Possible ways to incorporate these thermal-electro-chemical effects into a multi-phase LBM routine respectively are; ...”

In Section 2.5, extra details are added on the link between temperature, pressure, driving forces, and capillary numbers:

“To emulate the convective force of a minor temperature difference between the MPL and gas channel, an upwards body force of 10^{-6} (lattice units) was applied - similar to previous water management studies that impose pressure or velocity conditions in lieu of explicitly modeling temperature fields [12, 14]. This assumes that the effect of temperature gradient and any other upwards diving forces generated by density differences are relatively homogeneous, and in this case, results in a capillary number of 10^{-5} in the upwards direction.”

References

- [1] Multiple-Relaxation-Time Lattice Boltzmann Models in Three Dimensions. *Philosophical Transactions: Mathematical, Physical and Engineering Sciences*, 360:437–451, 2002.
- [2] R. T. Armstrong, J. E. McClure, M. A. Berrill, M. Rücker, S. Schlüter, and S. Berg. Beyond Darcy’s law: The role of phase topology and ganglion dynamics for two-fluid flow. *Physical Review E*, 94(4):043113, 2016.
- [3] S. R. Charlton and D. L. Parkhurst. Modules based on the geochemical model phreeqc for use in scripting and programming languages. *Computers & Geosciences*, 37:1653–1663, 2011.
- [4] P. Chaurand, W. Liu, D. Borschneck, C. Levard, M. Auffan, E. Paul, B. Collin, I. Kieffer, S. Lanone, J. Rose, et al. Multi-scale x-ray computed tomography to detect and localize metal-based nanomaterials in lung tissues of in vivo exposed mice. *Scientific reports*, 8(1):1–11, 2018.

- [5] S. Chen, X. He, V. Bertola, and M. Wang. Electro-osmosis of non-Newtonian fluids in porous media using lattice Poisson–Boltzmann method. *Journal of Colloid and Interface Science*, 436:186–193, 2014.
- [6] R. N. Chityala, K. R. Hoffmann, D. R. Bednarek, and S. Rudin. Region of interest (roi) computed tomography. In *Medical Imaging 2004: Physics of Medical Imaging*, volume 5368, pages 534–541. SPIE, 2004.
- [7] J. Cho, T. Neville, P. Trogadas, Q. Meyer, Y. Wu, R. Ziesche, P. Boillat, M. Cochet, V. Manzi-Orezzoli, P. Shearing, D. Brett, and M.-O. Coppens. Visualization of liquid water in a lung-inspired flow-field based polymer electrolyte membrane fuel cell via neutron radiography. *Energy*, 170:14–21, 2019.
- [8] M. Fan, L. E. Dalton, J. McClure, N. Ripepi, E. Westman, D. Crandall, and C. Chen. Comprehensive study of the interactions between the critical dimensionless numbers associated with multiphase flow in 3d porous media. *Fuel*, 252:522–533, 2019.
- [9] M. Fan, J. E. McClure, R. T. Armstrong, M. Shabaninejad, L. E. Dalton, D. Crandall, and C. Chen. Influence of clay wettability alteration on relative permeability. *Geophysical Research Letters*, 47(18):e2020GL088545, 2020.
- [10] R. Guo, L. E. Dalton, M. Fan, J. McClure, L. Zeng, D. Crandall, and C. Chen. The role of the spatial heterogeneity and correlation length of surface wettability on two-phase flow in a CO₂-water-rock system. *Advances in Water Resources*, 146:103763, 2020.
- [11] B. Han and H. Meng. Numerical studies of interfacial phenomena in liquid water transport in polymer electrolyte membrane fuel cells using the lattice boltzmann method. *International Journal of Hydrogen Energy*, 38(12):5053–5059, 2013.
- [12] F. Jinuntuya, M. Whiteley, R. Chen, and A. Fly. The effects of gas diffusion layers structure on water transportation using x-ray computed tomography based lattice boltzmann method. *Journal of Power Sources*, 378:53–65, 2018.
- [13] M. Junk, A. Klar, and L.-S. Luo. Asymptotic analysis of the lattice boltzmann equation. *Journal of Computational Physics*, 210(2):676–704, 2005.
- [14] K. N. Kim, J. H. Kang, S. G. Lee, J. H. Nam, and C.-J. Kim. Lattice boltzmann simulation of liquid water transport in microporous and gas diffusion layers of polymer electrolyte membrane fuel cells. *Journal of Power Sources*, 278:703–717, 2015.
- [15] B. Lim, S. Son, H. Kim, S. Nah, and K. M. Lee. Enhanced deep residual networks for single image super-resolution, 2017.
- [16] J. Liu, S. Shin, and S. Um. Comprehensive statistical analysis of heterogeneous transport characteristics in multifunctional porous gas diffusion layers using lattice boltzmann method for fuel cell applications. *Renewable Energy*, 139:279–291, 2019.
- [17] J. E. McClure, R. T. Armstrong, M. A. Berrill, S. Schlüter, S. Berg, W. G. Gray, and C. T. Miller. Geometric state function for two-fluid flow in porous media. *Physical Review Fluids*, 3(8):084306, 2018.
- [18] J. E. McClure, Z. Li, M. Berrill, and T. Ramstad. The LBPM software package for simulating multiphase flow on digital images of porous rocks. *Computational Geosciences*, pages 1–25, 2021.
- [19] Q. Meyer, S. Ashton, P. Boillat, M. Cochet, E. Engebretsen, D. P. Finegan, X. Lu, J. J. Bailey, N. Mansor, R. Abdulaziz, O. O. Taiwo, R. Jervis, S. Torija, P. Benson, S. Foster, P. Adcock, P. R. Shearing, and D. J. Brett. Effect of gas diffusion layer properties on water distribution across air-cooled, open-cathode polymer electrolyte fuel cells: A combined ex-situ x-ray tomography and in-operando neutron imaging study. *Electrochimica Acta*, 211:478–487, 2016.
- [20] Y. Niu, Y. D. Wang, P. Mostaghimi, P. Swietojanski, and R. T. Armstrong. An innovative application of generative adversarial networks for physically accurate rock images with an unprecedented field of view. *Geophysical Research Letters*, 47(23):e2020GL089029, 2020. e2020GL089029 2020GL089029.

- [21] W. Preeti and L. Smriti. Gpu as a service (gpuaas) market size by product: 2022 - 2030. *Global Market Insights*, 2022.
- [22] P. Rama, Y. Liu, R. Chen, H. Ostadi, K. Jiang, X. Zhang, R. Fisher, and M. Jeschke. An X-Ray Tomography Based Lattice Boltzmann Simulation Study on Gas Diffusion Layers of Polymer Electrolyte Fuel Cells. *Journal of Fuel Cell Science and Technology*, 7(3), 03 2010. 031015.
- [23] S. Sakaida, Y. Tabe, and T. Chikahisa. Large scale simulation of liquid water transport in a gas diffusion layer of polymer electrolyte membrane fuel cells using the lattice boltzmann method. *Journal of Power Sources*, 361:133–143, 2017.
- [24] P. Sarkezi-Selsky, H. Schmies, A. Kube, A. Latz, and T. Jahnke. Lattice boltzmann simulation of liquid water transport in gas diffusion layers of proton exchange membrane fuel cells: Parametric studies on capillary hysteresis. *Journal of Power Sources*, 535:231381, 2022.
- [25] P. Satjaritanun, F. C. Cetinbas, S. Hirano, I. V. Zenyuk, R. K. Ahluwalia, and S. Shimpalee. Hybrid lattice boltzmann agglomeration method for modeling transport phenomena in polymer electrolyte membrane fuel cells. *Journal of The Electrochemical Society*, 168(4):044508, apr 2021.
- [26] P. Satjaritanun, S. Hirano, I. V. Zenyuk, J. W. Weidner, N. Tippayawong, and S. Shimpalee. Numerical study of electrochemical kinetics and mass transport inside nano-structural catalyst layer of PEMFC using lattice boltzmann agglomeration method. *Journal of The Electrochemical Society*, 167(1):013516, oct 2019.
- [27] S. Shimpalee, P. Satjaritanun, S. Hirano, N. Tippayawong, and J. W. Weidner. Multiscale modeling of PEMFC using co-simulation approach. *Journal of The Electrochemical Society*, 166(8):F534–F543, 2019.
- [28] T. Spanos, V. De La Cruz, J. Hube, and R. Sharma. An analysis of buckley-leverett theory. *Journal of Canadian Petroleum Technology*, 25(01), 1986.
- [29] K. Tang, Q. Meyer, R. White, R. T. Armstrong, P. Mostaghimi, Y. Da Wang, S. Liu, C. Zhao, K. Regenauer-Lieb, and P. K. M. Tung. Deep learning for full-feature x-ray microcomputed tomography segmentation of proton electron membrane fuel cells. *Computers & Chemical Engineering*, 161:107768, 2022.
- [30] K. Tang, Y. D. Wang, P. Mostaghimi, M. Knackstedt, C. Hargrave, and R. T. Armstrong. Deep convolutional neural network for 3d mineral identification and liberation analysis. *Minerals Engineering*, 183:107592, 2022.
- [31] J. TÅ¶lke, G. D. Prisco, and Y. Mu. A lattice boltzmann method for immiscible two-phase stokes flow with a local collision operator. *Computers & Mathematics with Applications*, 65(6):864–881, 2013. Mesoscopic Methods in Engineering and Science.
- [32] J. Wang, M. Wang, and Z. Li. Lattice Poisson–Boltzmann simulations of electro-osmotic flows in microchannels. *Journal of Colloid and Interface Science*, 296(2):729–736, 2006.
- [33] J. Wang, M. Wang, and Z. Li. Lattice evolution solution for the nonlinear Poisson-Boltzmann equation in confined domains. *Communications in Nonlinear Science and Numerical Simulation*, 13(3):575–583, 2008.
- [34] M. Wang and Q. Kang. Modeling electrokinetic flows in microchannels using coupled lattice Boltzmann methods. *Journal of Computational Physics*, 229(3):728–744, 2010.
- [35] Y. Wang, T. Chung, R. Armstrong, J. McClure, T. Ramstad, and P. Mostaghimi. Accelerated computation of relative permeability by coupled morphological and direct multiphase flow simulation. *Journal of Computational Physics*, 401:108966, 2020.
- [36] Y. D. Wang, M. J. Blunt, R. T. Armstrong, and P. Mostaghimi. Deep learning in pore scale imaging and modeling. *Earth-Science Reviews*, 215:103555, 2021.
- [37] Y. D. Wang, T. Chung, A. Rabbani, R. T. Armstrong, and P. Mostaghimi. Fast direct flow simulation in porous media by coupling with pore network and laplace models. *Advances in Water Resources*, 150:103883, 2021.

- [38] A. Z. Weber, R. L. Borup, R. M. Darling, P. K. Das, T. J. Dursch, W. Gu, D. Harvey, A. Kusoglu, S. Litster, M. M. Mench, R. Mukundan, J. P. Owejan, J. G. Pharoah, M. Secanell, and I. V. Zenyuk. A critical review of modeling transport phenomena in polymer-electrolyte fuel cells. *Journal of The Electrochemical Society*, 161(12):F1254–F1299, 2014.
- [39] H. Zhang, L. Osmieri, J. H. Park, H. T. Chung, D. A. Cullen, K. C. Neyerlin, D. J. Myers, and P. Zelenay. Standardized protocols for evaluating platinum group metal-free oxygen reduction reaction electrocatalysts in polymer electrolyte fuel cells. *Nature Catalysis*, 5(5):455–462, 2022.
- [40] L. Zhang and M. Wang. Electro-osmosis in inhomogeneously charged microporous media by pore-scale modeling. *Journal of Colloid and Interface Science*, 486:219–231, 2017.

REVIEWER COMMENTS

Reviewer #2 (Remarks to the Author):

Thank you for the revised manuscript. All points have been addressed.

Reviewer #3 (Remarks to the Author):

To the reviewer's opinion, the authors addressed all the major aspects and provided a clear response to the reviewers comments.

There is one point left that I think was not correctly managed, regarding the sentence: "An air stoichiometry of 9 was set on the surface of the MPL to reproduce a differential cell configuration [80], as described in Section 4.5, whereby the gas/liquid fraction boundary condition is set to 9 on the MPL surface."

In the reply I do not see any clarification to the point, so I think the authors did not understand my concern and I try to clarify it more in detail. Air stoichiometry refers to the air flow rate that is fed to the inlet channel. A value of 9, I agree, is consistent with a differential flow configuration, and it means that the molar air flow rate that is fed to the channel inlet is very high and equal to: $9 \cdot I/4/F/y_{O_2}$, where I is the overall current, F the Faraday's constant, y_{O_2} oxygen mole fraction in air inlet. This boundary condition cannot be applied to the MPL (which is also not in contact with the channel) but to the air inlet section. In the MPL, the water generation flux is $I/A/2/F$ where A is the geometric area (under the assumption of uniform water generation according to the paper). I cannot understand how the arguments provided by the authors are consistent with the general definition of stoichiometry that is commonly used in the literature of the field, thus I kindly ask them to clarify this point. If this definition is correct they should provide the value of current (I) that is used in the simulation and clarify why it should consequently result that "the gas/liquid fraction boundary condition is set to 9 on the MPL surface". This last sentence misleads to think that the air stoichiometry is equal to the ratio between gas/liquid fractions at the MPL B.C. which is obviously not possible.

NCOMMS-22-32485 Response 2

December 2022

We thank reviewer 3 for their detailed comments and constructive suggestions to this manuscript. We have herein addressed all comments (in black) given to us, and **our responses and quotes from the original submission are in blue**, while our changes to the manuscript are in red.

Reviewer #3 (Remarks to the Author):

Comment #1, Reviewer #3

To the reviewer's opinion, the authors addressed all the major aspects and provided a clear response to the reviewers comments.

There is one point left that I think was not corretly managed, regarding the sentence: "An air stoichiometry of 9 was set on the surface of the MPL to reproduce a differential cell configuration [80], as described in Section 4.5, whereby the gas/liquid fraction boundary condition is set to 9 on the MPL surface."

In the reply I do not see any clarification to the point, so I think the authors did not understand my concern and I try to clarify it more in detail. Air stoichiometry refers to the air flow rate that is fed to the inlet channel. A value of 9, I agree, is consistent with a differential flow configuration, and it means that the molar air flow rate that is fed to the channel inlet is very high and equal to: $9 * I/4/F/y_{O_2}$, where I is the overall current, F the Faraday's constant, y_{O_2} oxygen mole fraction in air inlet. This boundary condition cannot be applied to the MPL (which is also not in contact with the channel) but to the air inlet section. In the MPL, the water generation flux is $I/A/2/F$ where A is the geometric area (under the assumption of uniform water generation according to the paper). I cannot understand how the arguments provided by the authors are consistent with the general definition of stoichiometry that is commonly used in the literature of the field, thus I kindly ask them to clarify this point. If this definition is correct they should provide the value of current (I) that is used in the simulation and clarify why it should consequently result that "the gas/liquid fraction boundary condition is set to 9 on the MPL surface". This last sentence misleads to think that the air stoichiometry is equal to the ratio between gas/liquid fractions at the MPL B.C. which is obviously not possible.

The reviewers comment regarding the air and water boundary conditions is apt, and the term stoichiometry, as suggested, should be removed and clarified with clearer boundary condition terminology. With the removal of the term stoichiometry, the details and assumptions made regarding the MPL and gas channel boundary conditions remain the same.

The simulation can be thought of as containing essentially 3 boundary conditions.

1. A flow rate of pure gas (air) is prescribed at the gas channel inlet.
2. An upwards pressure gradient is prescribed over the MPL and GDL to mimic the effects of temperature gradients.
3. The surface of the MPL is set to always contain a layer of 90% air and 10% liquid water by volume. This surface layer mimics the expected saturation that would emerge from within the MPL under the driving forces present (gas channel flow and upward forces). This ratio of 9:1 air to water assumes that (i) the MPL is homogeneous (excluding resolved fractures), (ii) the flow of water and air within the MPL follows Buckley-Leverett [1] two-phase flow, and (iii), that the amount of condensed liquid water within the MPL is relatively small compared to the flow of vapour - that is - it is not being flooded. This ratio can be modified to mimic the water-air dynamics of an MPL with a different relative permeability or operating condition, such as near flooding conditions approaching a 1:1 ratio or higher.

As mentioned in revision 1, the details on how the MPL surface is assigned is found in the Results Section 2.5 and Methods Section 4.5.

We have modified these sections to not erroneously refer to the term stoichiometry, rather, we add some extra details on the volume fraction assumptions:

Modified Section 2.5

The simulation is performed with the injection of air (vapour) through the PEMFC gas channels with a Reynolds number of 1. An air to liquid water volumetric ratio of 9:1 was set on the surface of the MPL as described in Section 4.5 to mimic non-flooded Buckley-Leverett [1] displacement of condensed water within the MPL.

Modified Section 4.5

The multi-phase LBM in this study is implemented on a two-phase immiscible system. Flow in the gas channel and GDL is modeled, while the MPL is modeled as a hydrophobic surface that generates both water and gas at a constant, uniformly distributed gas-liquid saturation. ... This treatment thus assumes that (i) the generation of water in the catalyst layer is homogeneous, and will flow homogeneously to the surface of the MPL, thus assuming the MPL (excluding resolved fractures) is a homogeneous porous media that follows Buckley-Leverett [1] two-phase flow, and (ii) the water generated emerges fully condensed along the surface of the MPL and the gas phase contains no information regarding the moisture content. The MPL surface saturation value can be set to mimic operating conditions whereby the amount of condensed liquid water within the MPL is relatively small compared to the flow of vapour - that is - it is not being flooded, or mimic the water-air dynamics of an MPL with a different relative permeability or operating condition, such as near flooding conditions approaching a 1:1 ratio or higher.

References

- [1] S. Buckley and M. Leverett. Mechanism of Fluid Displacement in Sands. *Transactions of the AIME*, 146(01):107–116, 12 1942.

REVIEWERS' COMMENTS

Reviewer #3 (Remarks to the Author):

I thank the author the kind clarification and the fast response.

I suggest to accept the manuscript in its present form.